

# Vulnerable JavaScript functions detection using stacking of convolutional neural networks

Abdullah Sheneamer

Computer Science Department, Jazan University, Jazan, Saudi Arabia

## ABSTRACT

System security for web-based applications is paramount, and for the avoidance of possible cyberattacks it is important to detect vulnerable JavaScript functions. Developers and security analysts have long relied upon static analysis to investigate vulnerabilities and faults within programs. Static analysis tools are used for analyzing a program's source code and identifying sections of code that need to be further examined by a human analyst. This article suggests a new approach for identifying vulnerable code in JavaScript programs by using ensemble of convolutional neural networks (CNNs) models. These models use vulnerable information and code features to detect related vulnerable code. For identifying different vulnerabilities in JavaScript functions, an approach has been tested which involves the stacking of CNNs with misbalancing, random under sampler, and random over sampler. Our approach uses these CNNs to detect vulnerable code and improve upon current techniques' limitations. Previous research has introduced several approaches to identify vulnerable code in JavaScript programs, but often have their own limitations such as low accuracy rates and high false-positive or false-negative results. Our approach addresses this by using the power of convolutional neural networks and is proven to be highly effective in the detection of vulnerable functions that could be used by cybercriminals. The stacked CNN approach has an approximately 98% accuracy, proving its robustness and usability in real-world scenarios. To evaluate its efficacy, the proposed method is trained using publicly available JavaScript blocks, and the results are assessed using various performance metrics. The research offers a valuable insight into better ways to protect web-based applications and systems from potential threats, leading to a safer online environment for all.

## INTRODUCTION

JavaScript is a commonly employed tool on web pages to enhance their dynamic functionality (https://developer.mozilla.org/en-US/docs/Web/JavaScript). In a study conducted by *Bichhawat et al. (2014)*, it was discovered that JavaScript is utilized in more than 95% of websites for front-end web development. However, malicious actors

Corresponding author
Abdullah Sheneamer,
asheneamer@jazanu.edu.sa

```
const express = require('express');
const xssFilters = require('xss-filters');
const util = require('util');
const app = express();
app.get('/', (req, res) => {   const unsafeFirstname =
req.query.firstname;
const safeFirstname =
xssFilters.inHTMLData(unsafeFirstname);
res.send(util.format('<h1>Tom%s</h1>', safeFirstname)); });

app.listen(3000);
```

**Figure 1  XSS filters.**

```
const express = require('express');
const db = require('./db');

const router = express.Router();

router.get('/email', (req, res) => {
  db.query('SELECT email FROM users WHERE id = ' +
req.query.id);
    .then((record) => {
      // logical flow
      res.send(record[0]);
    })
});
```

**Figure 2  SQL injection.**

frequently take advantage of JavaScript's dynamic capabilities to infect users' computers and mobile devices (*Zhou & Evans, 2015*). According to the Internet Security Threat Report of 2018, one in ten assessed URLs was identified as dangerous, with one in sixteen being classified as highly alarming. A substantial portion of this harmful software uses JavaScript scripting. For instance, one form of JavaScript-based attack is FormJacking, which focuses on e-commerce websites' checkout pages to steal user information and credit card details. On average, FormJacking (*Tanaka & Kashima, 2019*) attacks compromised approximately 4,800 websites per month in 2018. Multiple malicious JavaScript-based attacks are in existence, including XSS, drive-by download assaults, and distributed denial-of-service (DDoS) attacks (*Sachin & Chiplunkar, 2012*).

## Most common JavaScript vulnerabilities

In website design, JavaScript is the most popular programming language. JavaScript is used on 95% of websites, and by 67% of professional developers, a Stack Overflow poll found (*Hollander, 2023*). Examples of most common JavaScript vulnerabilities are buffer overflows, format string vulnerabilities, XSS filters as shown in Fig. 1, SQL injection flaws as shown in Fig. 2 and HTTP secure cookies as shown in Fig. 3.

```
const express = require('express');
const session = require('express-session');
const SQLiteStore = require('connect-sqlite3')(session);
const util = require('util');

// express-session configuration
const sessionMiddleware = session({
    store: new SQLiteStore({
      table: 'sessions',
      db: 'sessions.db',
      dir: __dirname
    }),
    secret: 'H@rden y0ur c00k1e5',
     saveUninitialized: false,
      resave: false,
      rolling: true,
      name: 'ssid',
      domain: 'localhost',
      httpOnly: true,
      secure: true,
      sameSite: 'strict'
});

const app = express();

// tell Express to use the 'sessionMiddleware'
app.use(sessionMiddleware);

app.get('/', (req, res) => {
  // trigger the 'Set-Cookie' (otherwise no cookie would be set)
  req.session.counter = (req.session.counter || 0) + 1;

  res.send(util.format('You have ve visited this page %dtimes',
    req.session.conter));
});

app.listen(4000, () => {
  console.log('Application listening on port 4000');
});
```

**Figure 3** HTTP secure cookies.

### Cross-site scripting

OWASP OWA (*Open Worldwide Application Security Project (OWASP), 2023*) identifies cross-site scripting (XSS) as a prevalent security concern in web applications. This type of attack occurs when a malicious actor injects malicious code into the client-side of an application. It often happens when an application incorporates data from web pages, whether unexpected or user-provided, without the necessary filtration or verification. For instance, one can employ XSS filters in an express application *via* NPM packages.

### The basic framework of JavaScript engine vulnerability detection

Incorrect logical reasoning by developers in the design and implementation phase increases the security risk due to the subsequent similarities between the mechanisms of JavaScript engine vulnerabilities and conventional software vulnerabilities. Although the JavaScript

engine operates as a high-level language interpretation tool, there are differences between this and typical applications since it adheres to particular language traits which have the abilities of custom JavaScript functions to accept variable parameter types. To improve performance *Shrivastava et al. (2016)*, self-modification of the virtual function table pointer within a specific class can be used during execution, as well as optimizing JIT (Just-In-Time) code. Current methods to detect JavaScript engine vulnerabilities fall into two categories. The first relies on static analysis to determine specific vulnerability types, while the second group uses fuzzing to dynamically reveal vulnerabilities. However, there are three main issues that require focus within the framework for detecting JavaScript engine vulnerabilities using both detection methods. These issues include the selection of the detection stage, the creation of detection samples, and the delicate between efficiency and accuracy. Researchers must invest much time and effort to understand the way in which these issues influence and constrain each other.

*Selection of the detection stage.* Within the static system analysis process, selecting the detection phase is an important stage. The execution phases of the JavaScript engine includes primarily input gathering, code compilation to produce byte code, optimization, interpretation, and execution. Complementary approaches are required to identify any possible issues for JavaScript engine interpretation and execution sets. After the engine's on-the-fly compilation and byte code generation, the produced code is optimized and executed during the compilation process. The machine's efficiency and speed relies upon this phase. In this stage, because of the JavaScript language's high-level language features, the logic of optimizing the generated bytecode is complicated, and carries a high risk of problems.

*Detection sample generation.* In order to identify JavaScript engine vulnerabilities *via* fuzzing, generating detection samples is required. Effective sample generation improves the likelihood of running the application in a unique state. Simple black-box testing involves generating independent instances with random data, a technique first developed in 1981 (*Ntafos, 1981*). One downside of black-box fuzzing is that, because of the randomness of the inputs, the program executes primarily within the code interval that handles the provided information. Conversely, the QuickFuzz (*Grieco, Ceresa & Buiras, 2016*) framework verifies sample validity using a grammar check tool prior to incorporating them into the detection process, ensuring that the corpus is accurately constructed. PEACH (*Eddington, 2011*) customizes PeachPit files in accordance with the tested program's sample input, partitioning the fields and contents that necessitate random generation.

*Efficiency and accuracy.* When using static vulnerability detection tools that are targeted towards the engine code, an experienced security investigator will spend a large amount of time reading and comprehending the code. This process requires unique insight into the execution flow of the entire engine and potential vulnerabilities. The quality of vulnerability detection by using fuzz testing is reliant upon efficiency and accuracy as key indicators of system quality. In the JavaScript engine, the input is usually received by reading a text file from a disc. However, disc operational efficiency for reading and

writing files is low compared to memory operations. Therefore, it is essential to effectively manage disc operations so that fuzzing efficiency is improved. In the target code, LibFuzzer (*Serebryany, 2016*) includes a sanitization function as a startup mechanism, which facilitates parallelization of fuzzing testing by generating multiple instances of fuzzing tests operating in their respective levels. To improve fuzzing instance duplications, Wen Xu introduced a unique system call mechanism based on AFL and LibFuzzer (*Xu et al., 2017*) and a dual file system to utilize the speed of a RAM-based temporary file system, resulting in productivity increases. To improve efficiency, many commercial organizations implement their fuzzing frameworks on massively parallelized clusters, examples of which include prominent examples including OSS-Fuzzer (*Mike Aizatsky & Whittaker, 2023*) and Google's Cluster-Fuzzer (*Google, 2023*). We suggest a unique method to identify vulnerabilities in JavaScript programs by using a Stacking of Convolutional Neural Networks models, including VGG16, VGG19, AlexNet, ResNet and LSTM. This methodology is applied to the three datasets: the Snyk platform (*Viszkok, Hegedus & Ferenc, 2021*), the Node Security Project (*Ferenc et al., 2019*), and the Apache Tomcat dataset (*Apache Software Foundation, 2023*). This approach detects related vulnerable code based on both vulnerable information and code features.

## An outline of vulnerabilities contained within JavaScript

Web browsers on the most widely used operating systems such as Windows, Linux, Android, and iOS often suffer from vulnerabilities within their JavaScript engines (*Alfadel et al., 2021*). Recently, this has increased with the discovery of several high-severity vulnerabilities in these engines. In 2016, 27 high-severity vulnerabilities were found, followed by 69 in 2017, and 55 in 2018. These critical flaws can compromise the engine's previously secure memory state. Exploiting these vulnerabilities allows the attacker to manipulate data and gain control of the program's execution flow, potentially allowing them to take over the entire system. Because many software applications now integrate JavaScript engines, network information system security is threatened by these vulnerabilities. JavaScript engine vulnerabilities are known for their intricate analysis, debugging challenges, and the devastating impact they can have in cyberattacks. Identifying JavaScript engine vulnerabilities can be addressed *via* two methods: static code analysis and dynamic program performance. Static code analysis examines and assesses the logical issues often found in JavaScript engine code. The dynamic program execution detection method detects unusual program behavior by changing the program input to reveal loopholes.

### Architecture of JavaScript engines

JavaScript engines operate in real-time and are written in languages such as C/C++, for example, with multiple high-level language interpreters. Several of the applications that support JavaScript language interfaces, such as Adobe Reader, Node.js, PDFium, and Windows Defender, also carry JavaScript engines and are also used in web browsers. A JavaScript engine is comprised of three components: a compilation framework, a virtual machine framework, and a runtime environment (Fig. 4). The virtual machine controls variables in the language, the runtime environment provides runtime services, and the compilation framework converts JavaScript code from a high-level language into byte code

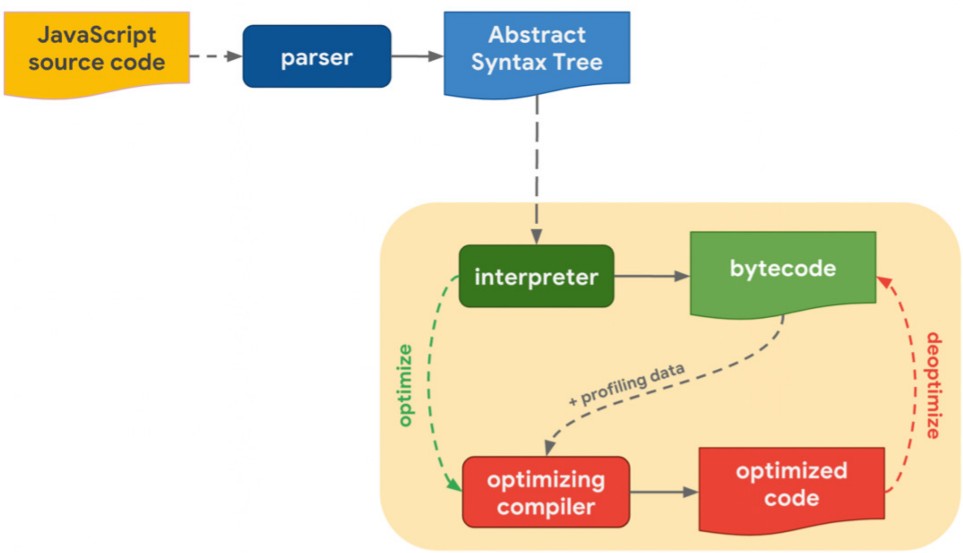

**Figure 4** **JavaScript engine architecture.**

or machine-native instructions which are then directly executed on the CPU (*Decan, Mens & Constantinou, 2018*).

### JavaScript engine type information

When the program executes, the engine uses type information and variable space. The prototype object, which the 'type' variable refers to, is preserved in the structure, and is based on the prototype inheritance. This object can be compared to a property collection that is comprised of key-value pairs, which are located in the same memory space and store data addresses, offset positions, or pointers to the object's location. An item's components are usually stored linearly in memory. For instance, 1.array =[];2. array[0] ="value1"; 3. Array [100] ="value100". For each line of code, the array is stored in a sparse mode. These arrays only require a few index values when accessing specific indexed items. So instead, they use an additional mapping in the index backing store to locate the portion at the relevant subscript. In addition to these storage structures, there are collections that can be optimized for data storage. To conserve memory, a set of 32-bit integers can be retained in their native form. The JavaScript engine defines various index types to differentiate between different types of array items stored in memory (*Smailbegovic, Gaydadjiev & Vassiliadis, 2005*).

### Vulnerability detection in JavaScript engines

Central to JavaScript engine vulnerability detection is the searching of input spaces of user-mode applications for an input that is capable of activating the vulnerabilities in a program. When the input has been received, the detection program records the read/write actions to memory, in addition to the engine's protective system during execution. The

ASAN (Address-Sanitizer) is usually added during compilation. During program execution, this compiler detects and adds in-time faults in memory operations. After an exception by the crash catcher is received (*Hallaraker & Vigna, 2005*), the input data is saved. Next, the engine separates its vulnerability identification into two categories: static analysis approach and fuzzing method. The static analysis method is characterized by accuracy and the basic vulnerability generation process. However, the evaluation needs significant quantities of energy and is inefficient. The fuzzing method can better focus on automating the actual process and minimizing the need for manual analysis (*Xu, Zhang & Zhu, 2013*; *Takanen et al., 2018*). This article's central contributions are:

- To provide a new stacking formal model that incorporates CNN architectures for the identification of JavaScript vulnerability functions.
- To perform vulnerability functions by investigating the application of static and process metric characteristics from JavaScript programs to CNN architectural models. The is the first known attempt to determine vulnerabilities by leveraging software that uses stacking of CNNs.
- To propose a multi-class system that combines various training and testing datasets to find all types of vulnerabilities.
- To produce comprehensive comparative research that use modern categorization methods for assessing the proposed approach.

This article addresses the following research questions:

**RQ1:** Can insecure JavaScript functions be located with the use of machine learning algorithms?

**RQ2:** How effective are the various deep learning architectures at identifying vulnerabilities, compared to each other?

**RQ3:** How effective is stacking different deep learning architectures, compared to other deep learning models?

**RQ4:** When using stacked CNN architectures, how do different classes of susceptible JavaScript methods perform?

This article is structured as follows: 'Prior research' highlights the background of the study. 'Proposed Method' describes previous works. 'Experiments' proposes a new method for detecting vulnerability JavaScript functions. 'Discussion of challenges using deep learning' evaluates and compares our proposed method and reports the results. 'Discussion of challenges using deep learning' discusses challenges using deep learning. 'Threat to validity' presents the threat to validity. 'Conclusions' gives the conclusion of the research.

## PRIOR RESEARCH

The JavaScript engine is a critical part of the browser, and acts as a virtual machine that executes JavaScript scripts. It is known for its intricate functional design, robust logic, and substantial quantity of code. Developing code with a comprehensive security system is very challenging to engine engineers. Applications that use JavaScript engines can face runtime

errors or possible crashes that result from design flaws and incorrect implementation, whilst others might be vulnerable to malicious exploitation by remote attackers.

JavaScript vulnerability detection methods have previous focused mainly on NPM package metadata. *Alfadel et al. (2021)* analyzed 2,904 open-source JavaScript from GitHub projects that were Dependable members and found that 65.42% of pull requests for security-related code are usually authorized and integrated within a one-day time period. Comprehensive analysis has found seven main reasons why Dependable security pull requests were not combined, and these are primarily concerned with concurrent improvements to the relevant dependencies than with a Dependable's shortcomings.

*Chinthanet et al. (2021)* conducted an empirical analysis to ascertain the possibility of delays between the susceptible release and the repairing releases(package-side fixing releases). An initial analysis of 231 package side fixing releases of GitHub NPM projects revealed that up to 85.72% of the packaged commits are unrelated to a patch, so a fixing release is not often solely distributed.

Over a 6-year period, *Decan, Mens & Constantinou (2018)* empirically researched 400 vulnerability reports in the NPM dependent network, which has more than 610 k JavaScript packages. They explored how and when vulnerabilities are identified and addressed.

*Duan et al. (2020)* proposed a comparison approach to evaluate the security and functional capabilities for interpreted languages of package managers. To investigate registry misuse, for example static, metadata, and dynamic analysis, they used well established software analytic methods based on a qualitative evaluation.

*Zerouali et al. (2019)* investigated the impact of vulnerabilities in NPM JavaScript packages within Docker images. The analysis was based on 1,099 security reports gathered from NPM and 961 images from three official *Node.js* repository sites. To assess NPM user security, package dependencies were examined, as well as the maintainers responsible for these packages, and the disclosure of security vulnerabilities. *Zimmermann et al. (2019)* investigated the idea that third-party dependencies could potentially facilitate the execution of unsafe or vulnerable programs. Their research revealed that even a single package could wield a substantial influence on the ecosystem.

*Kluban, Mannan & Youssef (2022)* developed a model for identifying vulnerabilities in real-world applications by using methods such as textual similarity and weak pattern recognition. This framework was built using a substantial dataset comprised of 1,360 confirmed insecure JavaScript processes obtained from the VulnCode-DB project and the Snyk vulnerability database.

To identify vulnerable routines, *Ferenc et al. (2019)* evaluated static code metrics gathered from static analyzers, including the quantity of code lines, nesting level, code complexity, etc. The highest performance achievement reached 0.7, which was measured by the F1 score. The authors concluded that the results can be improved by combining additional metrics with static source code measurements. *Mosolygó et al. (2021)* designed a method for assessing the vulnerability of a function using vector representations of tokenized code lines.

*Jain, Tomar & Sahu (2012)* created a set of attacks including password capture, phishing, click-jacking, and cookie theft to understand the effect of vulnerable JavaScript code on

a web page. Therefore, to detect vulnerable JavaScript code, regular expression-based matching mechanisms have been developed.

*Song et al. (2020)* recommended using a deep learning-based method to detect possible malicious code in JavaScript functions, and developed conceptual slices and the Program Dependency Graph (PDG) which maintain detailed semantic content and rapidly translate it into vectors to extract semantic features from JavaScript programs. Previous research has identified JavaScript vulnerabilities, although earlier research on JavaScript uses NPM package metadata-focused vulnerability detection approaches at the package level. This article presents a unique method for detecting JavaScript program vulnerabilities by employing several convolutional neural network models to discover related vulnerable code based on vulnerable data and code attributes.

*Alazab et al. (2022)* proposed an IDS designed to identify obfuscated JavaScript. This IDS uses a combination of features and machine learning methods to effectively distinguish between harmless and malicious JavaScript code. It also provides a unique set of features to identify obfuscation in JavaScript, as obfuscation is a widely used way to circumnavigate the usual systems to detect malware.

The proposed method was tested on JavaScript obfuscation attacks, with promising results. The IDS based on the chosen aspects reached a 94% detection rate for malicious samples, and for benign samples it was 81% within a feature vector dimension of 60.

*Bajantri & Shariff (2023)* compared Code BERT-based embedding with other mainstream embedding approaches, including Word2Vec, Glove, and Fast Text. The findings reveal that in downstream vulnerability detection tasks Code BERT-based embedding outperforms other methods. To improve efficiency further and to help the model learn susceptible code patterns in C, it is recommended to include synthetic vulnerable functions and perform fine-tuning using both real-world and synthetic data. The evaluation results shows that the updated Code BERT model exceeds various state-of-the-art identification techniques on the datasets used in the study.

*Lin et al. (2023)* developed an approach known as VulEye, which is a Graph Neural Network vulnerability detection method that is designed for PHP applications. The objective of VulEye is to assist security researchers to rapidly identify vulnerabilities in PHP projects. This involves building a Program Dependence Graph (PDG) from the PHP source codes and then slicing the PDG with sensitive functions to create Sub-Dependence Graphs (SDGs). This facilitates the detection of vulnerabilities in PHP.

*Nilavarasan & Balachander (2023)* addressed the XSS vulnerability, also known as cross-site scripting, which is a significant flaw in modern internet applications. The aim is to detect web attacks, and the study therefore investigates using deep learning techniques, particularly convolutional neural networks (CNNs), which are beneficial for XSS classification as a result of their architecture, which needs less pre-processing for extracting features. Here, the CNN method was used for categorizing and identifying XSS scripts as either benign or malicious, using the XSS script characters to create features. The results reveal that the respective values of accuracy, precision, and recall are 97.95%, 99.30%, and 96.66% respectively. This research only detects XSS vulnerabilities in JavaScript.

*Liu et al. (2023)* introduced a network vulnerability detection model based on multi-feature fusion-based neural MFXSS, which identifies cross-site scripting (XSS) vulnerabilities in the JavaScript source code of websites. This approach involves combining the code control flow graph (CFG) and the abstract syntax tree (AST) to transform the input data into code and graph string structures. To extract and merge logical call features and contextual execution relationship features from the source code, the model uses convolutional neural network graphs, bidirectional recurrent neural networks, and weighted aggregation. These combined feature vectors are then used for detecting and predicting JavaScript XSS vulnerabilities. The experiment included the design of multiple control experiments to optimize construction of the model, resulting in accuracy rates of 99.7% and 98.6% in standard and variant datasets, respectively. This research only detects XSS vulnerabilities within JavaScript.

*Chen et al. (2023)* introduced a vulnerable source code dataset, which was built by crawling security issue websites, extracting vulnerability-fixing commits, and corresponding source codes from other projects. The dataset was comprised of 18,945 vulnerable functions across 150 Common Weakness Enumerations (CWEs) and 330,492 non-vulnerable functions which were extracted from 7,514 commits. This dataset covers more projects than all preceding datasets combined.

By combining the new and preceding datasets, the researchers analyzed the challenges and potential research avenues for using deep learning in software vulnerability detection. Eleven model architectures that belong to 4 families were examined, and deep learning was determined to not yet be ready for vulnerability detection; this is because of high false positive rates, low F1-scores, and difficulties in detecting complex CWEs. They note the issue of generalization when using deep learning models and recommend increasing the amount of training data, which might not improve performance but could improve the ability of the model to generalize to unseen projects.

*Cheshkov, Zadorozhny & Levichev (2023)* evaluated ChatGPT-3 models for detecting vulnerabilities in code. The was conducted on real-world datasets using multi-label and binary classifications focused on vulnerabilities in CWE. The decision to evaluate these models was based on their good performance on other code-based tasks, including understanding code at a high level and solving programming challenges. Yet, the researchers discovered that in code vulnerability detection the ChatGPT model performed no better than a dummy classifier for both the multi-label and binary classification tasks. This indicates that the ChatGPT model may be unsuitable for this task.

## PROPOSED METHOD

This article employs the benefits of a well-established deep-learning algorithm to determine JavaScript capabilities with potential security vulnerabilities. Figure 5 shows the proposed method , and is comprised of two key processes: dataset preprocessing and creating a deep learning model.

Stacking multiple *1D* CNN models like VGG16, VGG19, AlexNet, ResNet, and LSTM models in order to detect JavaScript features involves combining the deep learning model capabilities with analysis of JavaScript code.

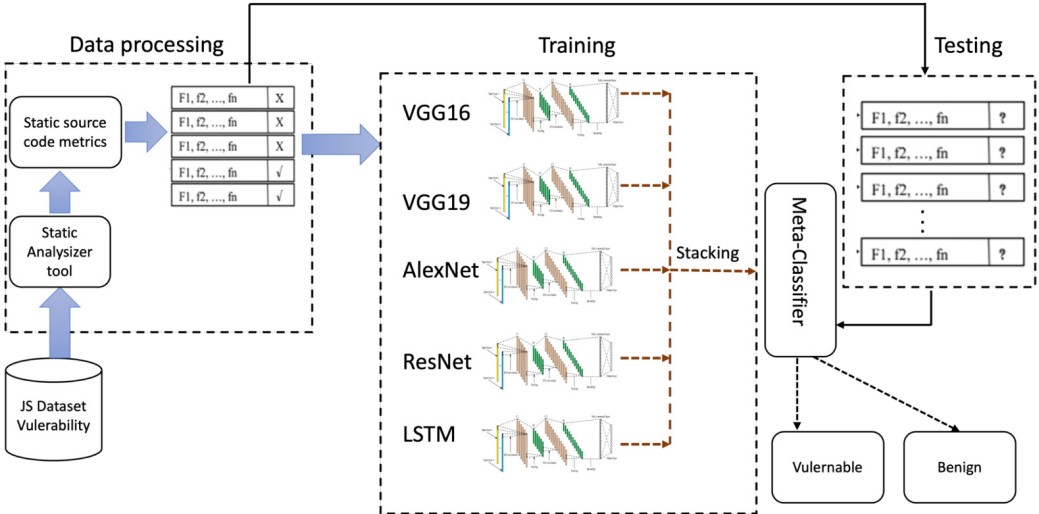

**Figure 5** Workflow of the proposed vulnerability performance prediction framework.

## A JavaScript vulnerability detection framework
### Data preparation
Two datasets in this study are from the Node Security Project (*Ferenc et al., 2019*) and the Snyk platform (*Viszkok, Hegedus & Ferenc, 2021*) with different source code metric sets, including static source code metrics and process source code metrics. The dataset and features were the same as those used by *Viszkok, Hegedus & Ferenc (2021)* and *Gyimesi (2017)*. To establish process metrics, they incorporated this technique into QualityGate, a software quality tracking tool that can rapidly browse through thousands of program versions. The platform a project's git version control as input data and performs a static evaluation, stores the results in an internal graph structure, and then computes process metrics. To calculate procedure metrics up to a specific commit hash, the versioning system URL for each project is required. The process metrics are then updated based on the previous values and the most recent contribution for each program revision. To determine the process metrics, an analysis of the original database is conducted, the required project information acquired, and QualityGate used to incorporate process metrics for each function into the original dataset.

### Extract features
The JavaScript code is tokenized into different tokens: words, symbols, and operators in the code. A parser is the used to generate an abstract syntax tree (AST) representation of the code. The tree captures the code's structural information. Static features are then extracted, as displayed in Tables 1 and 2. Next, the process source code features include dynamic attributes which require code execution to collect the relevant information, as shown in Table 3. Each code sample is executed within a controlled environment, with behavior closely monitored. This execution provides dynamic detail, such as APIs, data

flows, and runtime errors. The static and process features results are then combined into an individual feature vector for each code sample.

### Normalization and preprocessing

The static and process features vectors are then normalized to ensure their consistency on a scale. The dataset is then divided into training, validation, and testing sets.

### Preprocessing for CNN models

The feature vectors are resized to consistent input sizes as required by the models (*e.g.*, 224 × 224 pixels). The values are normalized to the range expected by the models (usually between 0 and 1).

### Training CNN models

We train each deep learning model (VGG16, VGG19, AlexNet, ResNet, and LSTM) using the prepared data. Each output layer is adjusted to match either the multi-classes or the binary classification task (vulnerable or non-vulnerable). Table 4 provide overviews of the hyperparameters and architectural details for *1D* versions of VGG16, VGG19, AlexNet, ResNet, and *1D* LSTM networks, which are usually employed for sequential data analysis.

### Stacked models

Similar to bagging and boosting methods, a stacked model integrates the predictions from the five models on the same dataset (VGG16, VGG19, AlexNet, ResNet, and LSTM). The complete architecture of the stacked model is shown in Fig. 5. The models at Level 0 (BaseModels), are built using predictions from models trained on the training data. There is a model at Level 1 that determines the optimum approach for combining base model predictions. This meta-model is trained using non-sample data from underlying model extrapolations. Thus, data that was not used to build the base models is introduced to them, predictions are generated which, with the corresponding projected outputs, form the input–output pairs of the training dataset used for training the meta-model. For regression, categorization, and classification-like models, the base model outputs are the input. This output adopts different forms, including actual data values, probabilities, probability-like values, or class labels. A dedicated testing dataset is used to evaluate the stacked model, and its performance is assessed based on static and process metrics. The best-performing model is used for real world JavaScript vulnerability detection, which increases web application security.

## Overview of deep learning models

Two deep-learning models were used to detect JavaScript function vulnerabilities: the VGG16 model, the VGG19 model, and the Alexnet model.

### VGG16 model

*Simonyan & Zisserman (2014)* and *Sherstinsky (2020)* at the University of Oxford introduced the VGG16 model, which is a CNN architecture. Even though the model's concept was first indicated in 2013, it was only in 2014 (*Thite, 2023*) that it was officially submitted for the ILSVRC ImageNet Challenge. VGG16 is a very effective model for

**Table 1  Static metrics (*Ferenc et al., 2019*; *Viszkok, Hegedus & Ferenc, 2021*).**

| Metric | Description |
| --- | --- |
| CC | Number of Clone Coverage |
| CCL | Number of Clone Classes |
| CCO | Number of Clone Complexity |
| CI | Number of Clone Instances |
| CLC | Number of Clone Line Coverage |
| LDC | Number of Lines of Duplicated Code |
| McCC and CYCL | Number of Cyclomatic Complexity |
| NL | Number of Nesting Level |
| NLE | Number of Nesting Level without else-if |
| CD | Number of Comment Density |
| TCD | Total Comment Density |
| CLOC | Number of Comment Lines of Code |
| TCLOC | Total Comment Lines of Code |
| DLOC | Number of Documentation Lines of Code |
| LLOC | Number of Logical Lines of Code |
| TLLOC | Total of Logical Lines of Code |
| LOC | Number of Lines of Code |
| TLOC | Total Lines of Code |
| NOS | Number of Statements |
| TNOS | Total Number of Statements |
| NUMPAR and PARAMS | Number of Parameters |
| HOR D | Number of Distinct Halstead Operators |
| HOR T | Number of Total Halstead Operators |
| HON D | Number of Distinct Halstead Operands |
| HON T | Number of Total Halstead Operands |
| HLEN | Halstead Length |
| HVOC | Halstead Vocabulary Size |
| HDIFF | Halstead Difficulty |
| HVOL | Halstead Volume |
| HEFF | Halstead Effort |
| HBUGS | Number of Halstead Bugs |
| HTIME | Halstead Time |
| CYCL_DENS | Cyclomatic Density |
| WarningInfo | ESLint Info priority Warnings |
| WarningMinor | ESLint Minor priority Warnings |
| WarningMajor | ESLint Major priority Warnings |
| WarningCritical | ESLint Critical priority Warnings |
| WarningBlocker | ESLint Blocker priority Warnings |

**Table 2  List of static source code metrics (*Ganesh, Palma & Olsson, 2022*).**

| Metric | Description |
| --- | --- |
| anonymousClassesQty | Number of anonymous classes in a class. |
| anonymousClassesQty | Number of anonymous classes in a class. |
| assignmentsQty | Number of times of use of each variable inside each class. |
| CBO | Coupling Between Objects. The number of dependencies in a class. |
| comparisonsQty | Represent the class's comparison operators. |
| defaultFieldsQty | Number of default field types. |
| DIT | Inheritance Tree Depth. Number of Parents in a class. |
| fieldQty | Number of use of each local field inside each class |
| finalFieldsQty | Number of fields of final types. |
| HasJavadoc | If the source code has JavaDoc or not. |
| innerClassesQty | Number of inner classes in a class. |
| lambdsQty | Number of lambda expressions in a class. |
| LCC | Loose Class Cohesion. |
| LCOM | Lack of Cohesion of Methods |
| LOC | Number of Lines of Code. |
| logStatementsQty | Count of log statements in a class. |
| loopQty | Number of loops in a class. |
| mathOperationsQty | Represent the number of arithmetic symbols in a class. |
| maxNestedBlocksQty | Max Nested Blocks. |
| Method Invocations | Number of directly invoked methods |
| NoSI | Number of Static Invocations. |
| numbersQty | Number of numeric literals in a class. |
| paranthesizedExpsQty | Number of parenthesized expressions in a class. |
| privateMethodsQty | If a class is private. |
| protectedFieldsQty | Number of fields of protected types. |
| protectedMethodsQty | If a class is protected. |
| returnQty | Number of return statements in a class. |
| RFC | Response for a Class. The number of unique method invocations in a class. |
| stringLiteralsQty | Number of string literals in a class. |
| synchronizedMethodsQty | Number of synchronized methods in a class. |
| TCC | Tight Class Cohesion. |
| totalFieldsQty | Number of total fields. |
| totalMethodsQty | Number of Methods. |
| tryCatchQty | Number of the try and catch statements used in a class. |
| uniqueWordsQty | Number of unique words in a class. |
| variablesQty | Number of variables in a class. |
| visibleMethodsQty | Number of Visible Methods. |
| WMC | Weight Method Class. |

**Table 3 Process metrics (*Viszkok, Hegedus & Ferenc, 2021*).**

| Metric | Description |
| --- | --- |
| AVGNOAL | Average Number of New Lines |
| AVGNODL | Average Number Of Omitted Lines |
| AVGNOEMT | Average Number Of Elements Modified Together |
| AVGNOML | Average Number of Changed Lines |
| AVGTBC | Average Time Between Changes |
| CChurn | Sum of lines added minus lines Omitted |
| MNOAL | Maximum Number of New Lines |
| MNODL | Maximum Number of Omitted Lines |
| MNOEMT | Maximum Number of Elements Changed Together |
| MNOML | Maximum Number of Changed Lines |
| NOADD | Number of Additions |
| NOCC | Number of Contributor Changes |
| NOCHG | Number of Changes |
| NOContr | Number of Contributors |
| NODEL | Number of Deletions |
| NOMOD | Number of Modifications |
| SOADD | Sum of New Lines |
| SODEL | Sum of Omitted Lines |
| SOMOD | Sum of Changed Lines |

**Table 4 Hyberparameters of CNN architectures.**

| Parameter | Value |
| --- | --- |
| Batch size | Dataset size |
| Number of Convolutional Layers | VGG16 has 13, VGG19 has 16, AlexNet 5 convlutional layers and ResNet can have a varying number of layers. LSTM uses sequential data. |
| Learning rate $\theta$ | 0.001 |
| Regularization srength $\beta$ | 0.001 |
| Input shape | 128 |
| epoch | 1,000 |
| Activation function | ReLU, softmax, and sigmoid |
| Momentum | 0.9 |
| Optimizer | SGD(lr=learning_rate, momentum=momentum) |
| Loss function | categorical_crossentropy and binary_crossentropy |

document classification. This study uses the VGG16 model by training it from scratch, and refers to training the model on the datasets that begin with randomly initialized weights. This approach necessitates a large quantity of labeled data and computational resources to train the model. In this study the model's output uses an MLP classifier after training holistic features with these trained weights. The final classification was obtained from the sigmoid classifier.

### VGG19 model

According to *Simonyan & Zisserman (2014)*, VGG19 is a variant of the VGG model and has 19 layers (16 convolution layers, three fully linked layers, five MaxPool layers, and one SoftMax layer). VGG19 was trained using the basic model and by training VGG19 from scratch. The model outputs were then merged in an MLP classifier to produce the final classification from the sigmoid classifier.

### Alexnet model

Alex Krizhevsky, Ilya Sutskever, and Geoffrey Hinton (*Gershgorn, 2017*) developed the CNN, an AlexNet architecture. The initial five layers of AlexNet were comprised of convolutional layers, and then by max-pooling layers, and the final 3 layers are fully connected layers. The network used the non-saturating ReLU activation function, which outperformed tanh and sigmoid in regard to training performance. To train the base model this research used trained weights from scratch with AlexNet. All of the model's outputs were then combined in an MLP classifier. A sigmoid classifier was used to obtain the final classification.

While AlexNet was mainly intended for image classification, it does have other uses, including vulnerability code analysis. However, coding vulnerability detection differs from image classification, and the architecture needs to be appropriately modified. To modify AlexNet for vulnerability code analysis, the input data has to be preprocessed to represent code snippets. The architecture must also be modified to consider programming language specifics and the sequential flow of code.

To improve its performance when analyzing vulnerability code, recurrent neural networks (RNNs) can be included in the design of AlexNet. RNNs can help to capture temporal dependencies in code and are can be used for sequential data. Moreover, convolutional neural networks can be used to extract features from the code snippets. Overall, AlexNet can be a starting point for vulnerability code analysis, but requires significant modification to be successful. Other architectures such as LSTM and GRU may have greater relevance to this type of analysis.

### Residual neural netowrk (ResNet) model

Residual neural networks (ResNets) (*Targ, Almeida & Lyman, 2016*) have widely used due to their ability to solve complex image classification tasks. However, they can also be used for other purposes, such as vulnerability code analysis whereby, ResNets can be trained to detect common coding mistakes that can lead to security vulnerabilities. The objective would be to create a ResNet that could accurately identify potential code vulnerabilities in order to help defeat cyber threats.

### Long short-term memory (LSTM) model

Long short-term memory (LSTM) is a recurrent neural network (RNN) architecture (*Graves, 2012*; *Sherstinsky, 2020*) often used in natural language processing (NLP) applications such as sentiment analysis and language translation. LSTM networks handle sequential data, so their use in time series research and forecasting is very useful.

**Table 5  Datasets details.**

| Dataset | Total of functions | Vulnerability | Non-vulnerability |
|---|---|---|---|
| Ferenc et al.'s dataset (*Ferenc et al., 2019*) | 12,125 | 1,496 | 10,629 |
| Viszkok et al.'s dataset (*Viszkok, Hegedus & Ferenc, 2021*) | 8,038 | 960 | 7,078 |
| Apache Tomcat Dataset (*Ganesh, Palma & Olsson, 2022*) | 10,204 | 1,719 | 8,485 |

LSTM models can detect patterns in code which may indicate vulnerabilities in terms of security code analysis. The model can categorize new code segments as either susceptible or nonvulnerable after training on a dataset of known vulnerabilities and non-vulnerabilities.

An advantage to using LSTM models for vulnerability code analysis is that they can handle long-term dependencies, so during classification the model can consider the context of the code snippet. This can produce more accurate results compared to traditional machine learning models that do not consider the data's sequential nature.

LSTM models have potential for vulnerability code analysis and can improve the efficacy of vulnerability detection and mitigation efforts.

# EXPERIMENTS

## Dataset description

This research used publicly available datasets, including the Apache Tomcat dataset (*Apache Software Foundation, 2023*), shown in Table 5, and datasets from the Node Security Project (*Ferenc et al., 2019*) and the Snyk platform (*Ferenc et al., 2019*). Figure 6 shows the distribution of Apache Tomcat vulnerabilities, with Denial of Service (*Nsrav, 2023 (accessed May 6, 2023*), Remote Code Execution (*Hurley et al., 2005*), and Information Disclosure (*Harley, 2022 (accessed July 14, 2022*) being prominent vulnerabilities. Figure 7 shows 4 severity levels: Low, Moderate, Important, and High. To analyze the impact of vulnerabilities, we used Java classes in Apache Tomcat that are affected by the vulnerabilities. The dataset is comprised of static source code metrics as predictors or features, together with an extension that incorporates 19 different process metrics, as shown in Table 3. The metrics for the functions were calculated and entered into the final dataset that utilize two tools: escomplex (*Sachin & Chiplunkar, 2012*) and OpenStaticAnalyzer (OSA). The calculated metrics are listed in Tables 1 and 3.

## Performance measures

We used the standard performance measure, which is popular in vulnerable JavaScript code detection. The measures are precision, recall, F1-score, and accuracy. The definition of precision is the number of correctly classified vulnerable JavaScript function out of total JavaScript functions extracted, and is therefore computed as

$$Precision = \frac{TP}{TP + FP}. \tag{1}$$

Here, TP is true positives, *i.e.*, the amount of JavaScript functions classified as vulnerability functions, while FP refers to false positives, *i.e.*, the number of non-vulnerability functions

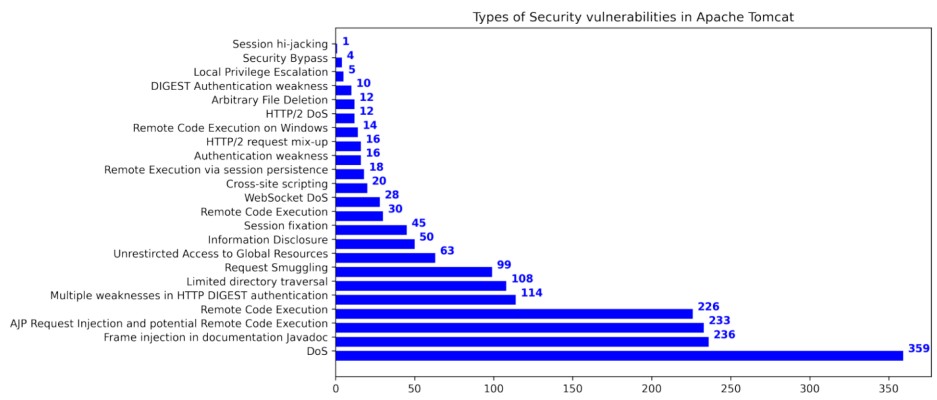

**Figure 6** Types of security vulnerabilities in Apache Tomcat (*Ganesh, Palma & Olsson, 2022*).

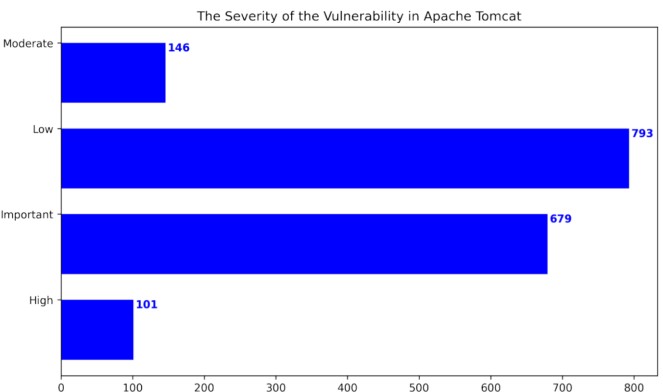

**Figure 7** The severity of the vulnerability in Apache Tomcat (*Ganesh, Palma & Olsson, 2022*).

determined as vulnerability functions. Recall is the ratio between predicted JavaScript functions that are actual vulnerability functions, and total true vulnerability functions:

$$Recall = \frac{TP}{TP + FN}. \tag{2}$$

Here, FN is False Negatives, *i.e.,* the number of vulnerability functions classified as non-vulnerability functions. TN means True Negatives, *i.e.,* the number of non-vulnerability functions determined as non-vulnerability functions. The f1- score combines the precision and recall:

$$f1-score = \frac{2 \times Precision \times Recall}{Precision + Recall}. \tag{3}$$

Finally, accuracy is the fraction of the vulnerability functions that are classified correctly:

$$Accuracy = \frac{TP + TN}{TP + TN + FN + FP}. \tag{4}$$

## Evaluation

### Research questions

This article address four research questions, alongside analysis of the results to support the findings using *Ferenc et al.'s (2019)* dataset and the Apache Tomcat dataset.

**RQ1. Can machine learning algorithms identify poor JavaScript functions? (A number of machine learning classifiers' potency**. This study examines how well various machine learning classifiers perform in the detection of software vulnerabilities, including Decision Tree (CART) (*Lewis, 2000*), logistic regression (LR) (*Hosmer Jr, Lemeshow & Sturdivant, 2013*), Naive Bayes (*Rish, 2001*), XGboost (*Chen & Guestrin, 2016*), Bagging (*Breiman, 1996*), Random Forest (RF) (*Breiman, 2001*), k-nearest neighbors (KNN) (*Fix, 1985*), support vector machine (SVM) (*Hearst et al., 1998*), and Stacking.

The performance of these classifiers across three datasets was assessed, with each comprised of three scenarios: an imbalanced case, an under-sampled case, and an oversampled case. For each dataset and scenario, comprehensive training and testing of multiple classifiers was carried out to evaluate their efficacy. To do this, a collection of code samples were categorized as either vulnerable or non-vulnerable. The creation of training and testing sets involved a random division of the dataset, with 67% allocated to the training set and 33% to the testing set for each of the three datasets and cases. After training the classifiers on the training set, their performance was assessed using the testing set, using standard evaluation metrics like precision, recall, and F1-score.

The results reveal that, across all three datasets and cases, some of the classifiers perform better in detecting code vulnerabilities. Specifically, it was discovered that the XGboost, Bagging, RF and stacking classifiers consistently outperformed the other classifiers. Regarding imbalance, the Cart and KNN classifiers performed comparably well as shown in Figs. 8, 9 and 10, and it was also observed that oversampling improves the performance of the classifiers, while under sampling can reduce performance as shown in Figs. 8, 9 and 10.

The Apache Tomcat dataset results show that of all the classifiers the Stacking classifier was the highest performing, with an f1-score of 99.58% and an accuracy of 99.58%. The RF classifier achieved an F1-score of 99.57% and an accuracy of 99.56%. The accuracies of the Bagging, XGboost, Cart, and KNN classifiers were 99.51%, 99.50%, 99.30%, and 97.39%, respectively, also show relatively high performance.

This study reveals the importance of choosing a relevant machine learning classifier to identify software vulnerability code. Moreover, the findings indicate that incorporating oversampling techniques improves classifier performance. It is recommended to use XGboost, Random Forest (RF), and stacking classifiers for software vulnerability code identification. However, to achieve optimal performance, other classifiers may require fine-tuning.

**RQ2: Compared to each other, how effective are the deep learning architectures at identifying vulnerabilities? (The Effectiveness of Various Deep Learning Architectures)** The performance of various CNN architectures was examined, including VGG16, VGG19, AlexNet, ResNet, and LSTM, in the detection of vulnerable code. To conduct these assessments, three distinct datasets were used, each featuring three scenarios of class

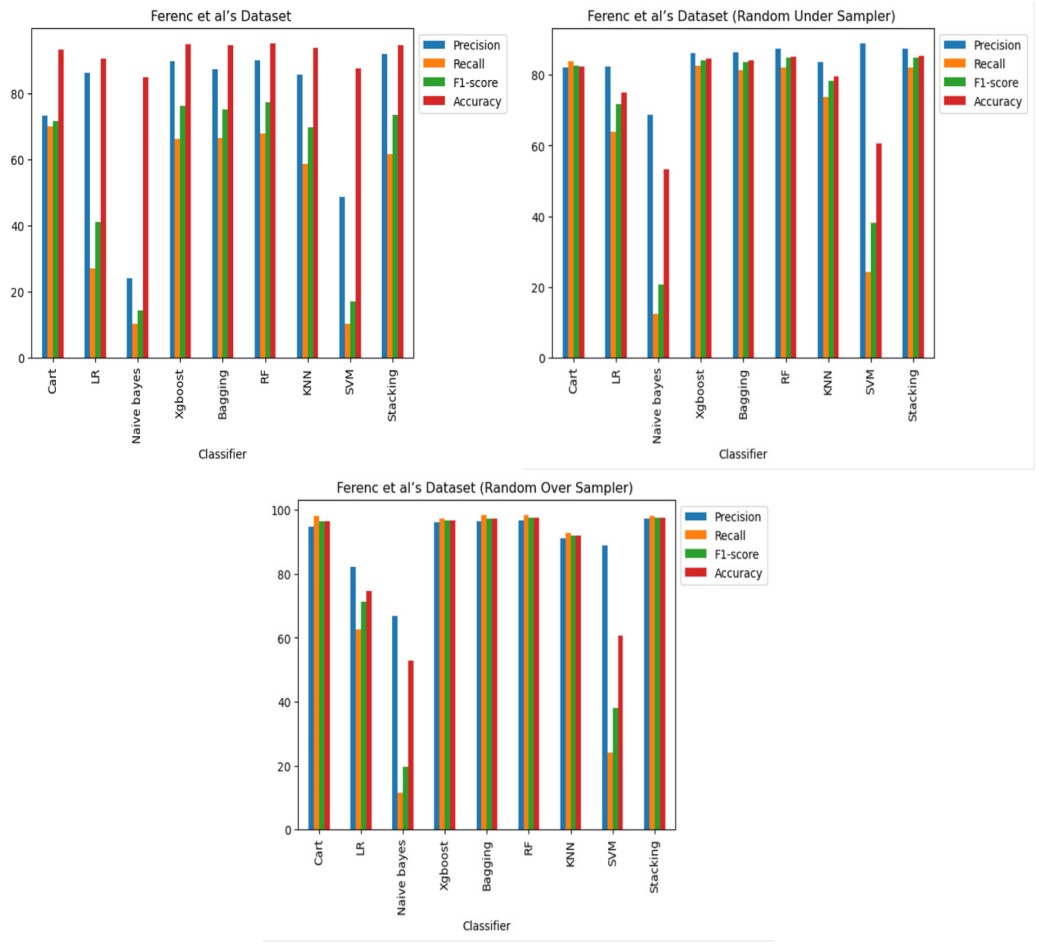

**Figure 8** *Ferenc et al.*'s (*2019*) dataset results using machine learning.

imbalance, random under-sampling, and random over-sampling. The datasets used were *Ferenc et al.*'s (*2019*) dataset, Viszkok et al.'s dataset, and the Apache Tomcat dataset, all of which are used widely in vulnerability code detection. For each dataset, the data was preprocessed to account for three cases of class imbalance, employing random under-sampling and random over-sampling techniques. This allowed an assessment of the effectiveness of CNN models in classifying vulnerable code. Subsequently, the CNN models were trained and tested using standard parameter settings specific to each dataset.

The findings indicate that in the context of vulnerability software detection, the Stacking and VGG16 architectures perform best. In particular, in *Ferenc et al.*'s (*2019*) dataset for the imbalanced dataset scenario, they achieved accuracies of 93.53% and 91.23%, as shown in Figs. 11 and 12. Conversely, in the random under-sampler dataset scenario, Stacking and the AlexNet architecture outperformed other models, achieving accuracies of 81.48% and 81.38%. In addition, for the random over-sampler dataset case in *Ferenc et al.*'s

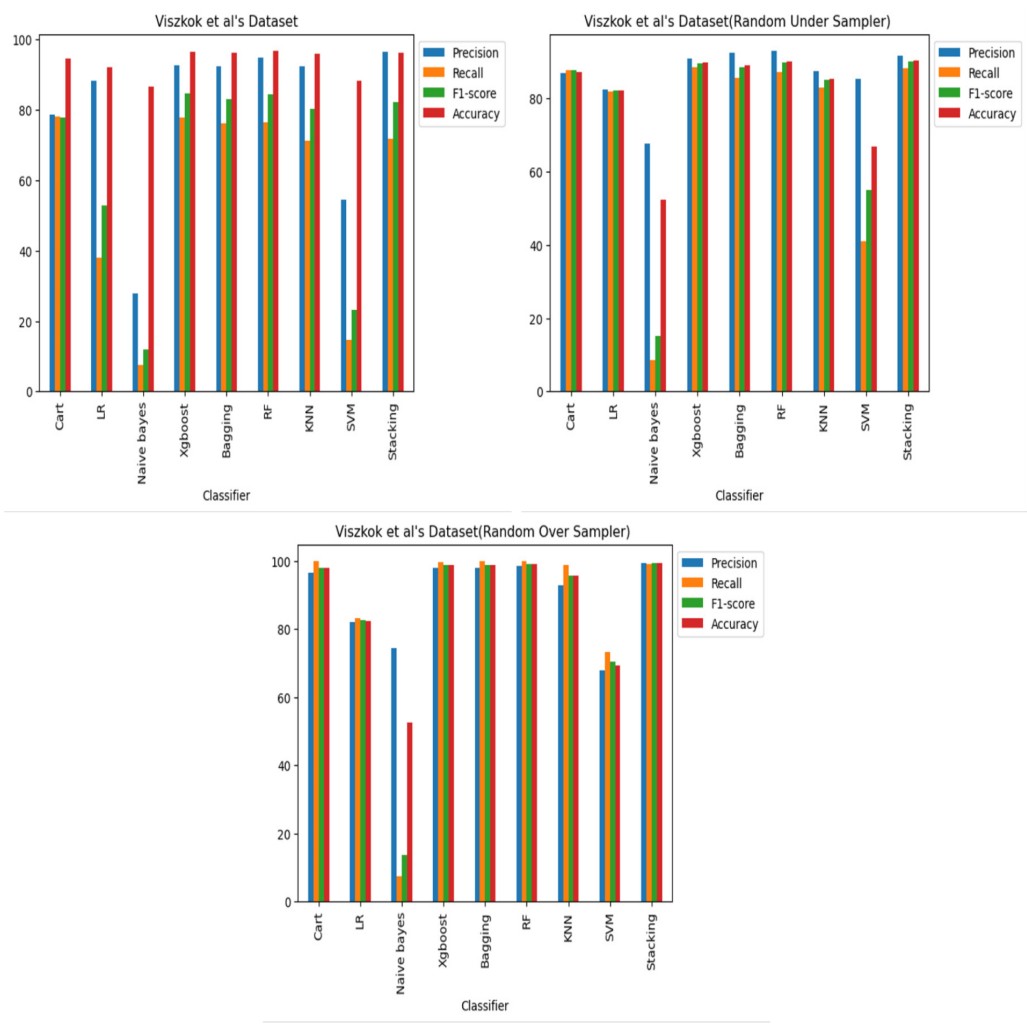

**Figure 9  Viszkok et al.'s dataset (*Viszkok, Hegedus & Ferenc, 2021*) results using machine learning.**

(*2019*) dataset, the Stacking and VGG19 architectures had better performance, achieving accuracies of 94.44% and 94.10%.

For Viszkok et al.'s dataset (*Viszkok, Hegedus & Ferenc, 2021*), as shown in Figs. 13 and 14, the VGG19 achieved 90.08% accuracy for the imbalance dataset case, 78.95% accuracy for the random under sampler cases, and 94.10% accuracy for the random over sampler cases. For Viszkok et al.'s dataset, the VGG19 achieved accuracy for imbalance dataset cases, accuracy for random under sampler cases, and accuracy for random over sampler cases.

For the Apache Tomcat dataset as shown in Figs. 15 and 16, the VGG19 architecture achieved 97% accuracy for imbalance dataset cases, 92.69% accuracy for random under sampler cases, and 98.80 accuracy for random over sampler cases. VGG16, AlexNet, ResNet, and LSTM models performed moderately well in our experimentation for specific cases of dataset pre-processing. It is recommended that on a complex dataset environment the

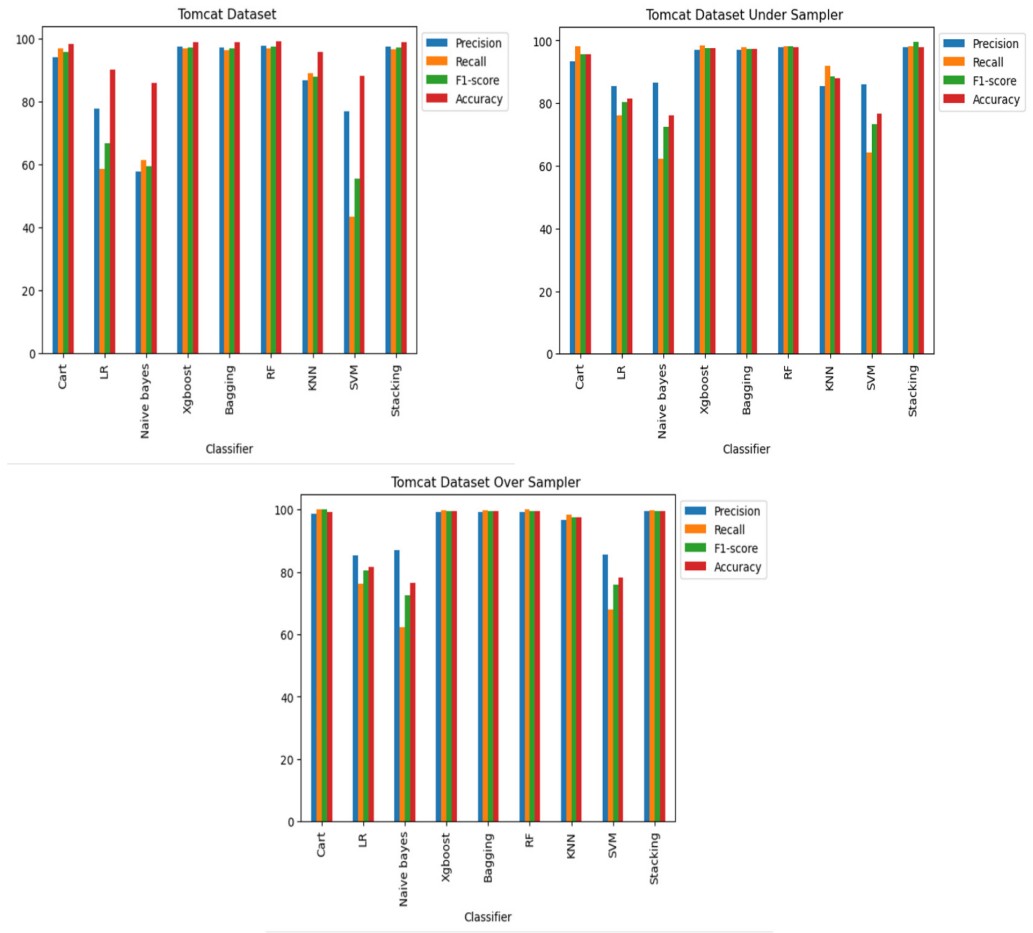

**Figure 10 Apache Tomcat dataset results using machine learning.**

VGG19 architecture should be used for vulnerability code detection, and models such as VGG16, AlexNet, ResNet, and LSTM for specific cases to achieve higher accuracy.

**RQ3: Compared to other deep learning architectures, how does the performance of stacking the various deep learning architectures? (Performance of Stacking of different CNN Architectures)** In order to evaluate performance, a comparative analysis was performed by running all CNN classifiers, including the stacked CNN architectures, on two separate occasions. The candidate models were trained, tested, and executed through data splitting, which involved dividing the dataset to construct the model and estimate its parameters in the learning phase. For model evaluation, a distinct and entirely new dataset was used, ensuring that the ratio between vulnerability and non-vulnerability classes stayed consistent with the overall dataset.

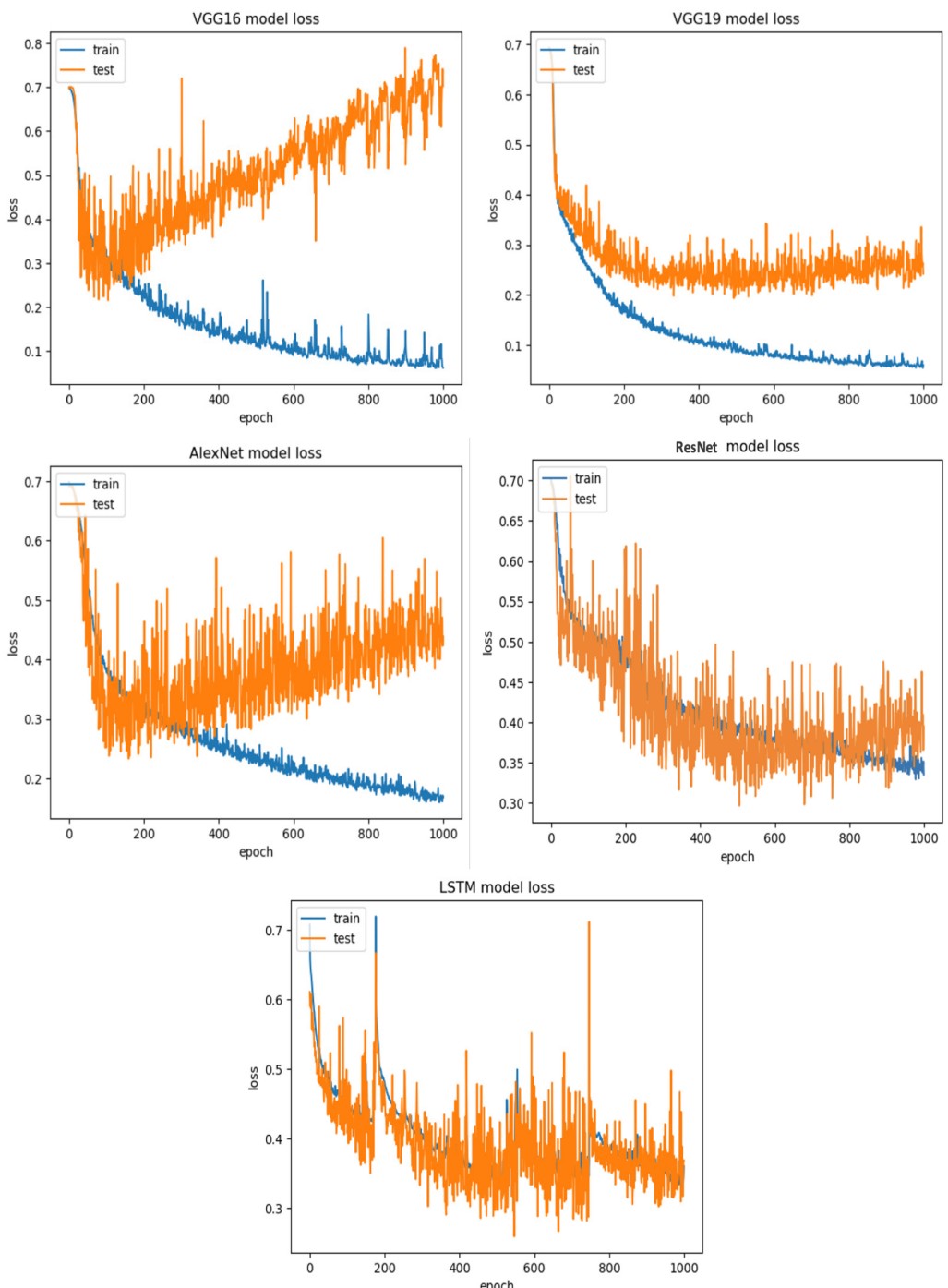

**Figure 11  Ferenc et al.'s (2019) dataset results using five deep learning architectures.**

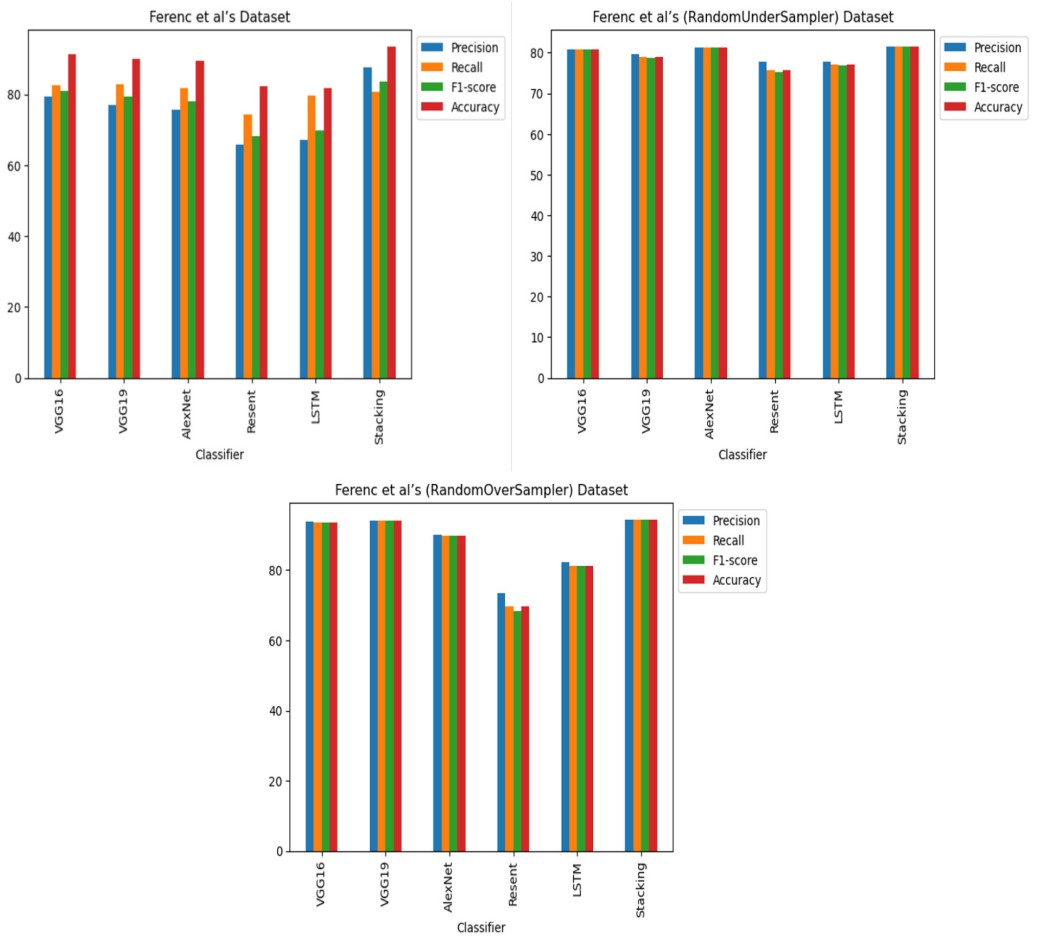

**Figure 12** *Ferenc et al.*'s (*2019*) **dataset results using deep learning architectures.**

Figures 12, 14 and 16 provide a comparison of the CNN classifiers using static and process metrics on the dataset. Amongst all the CNN classifiers, the results demonstrate that the stacked CNN architecture consistently achieves improved outcomes.

In terms of performance, it was found that stacking CNN architectures outperformed VGG16, VGG19, AlexNet, ResNet, and LSTM architectures. However, CNN performs surprisingly poorly in this scenario. Although 100 features were used in this study, it is possible that small dataset is the cause of the experiment's poor performance. Generally, the deep learning model automatically extracts features; in this scenario, feature extraction was carried out independently. This might contribute to the poor performance. However, in each of the six small datasets, XGboost performs significantly better.

**RQ4: How does CNN architecture stacking perform for multi-classes of vulnerable JavaScript functions? (Performance of Stacking of different CNN Architectures for various types of vulnerable JavaScript functions)**

Good performance of stacking different CNN architectures can be achieved for multi-class classification. This is the first known attempt to try and detect different

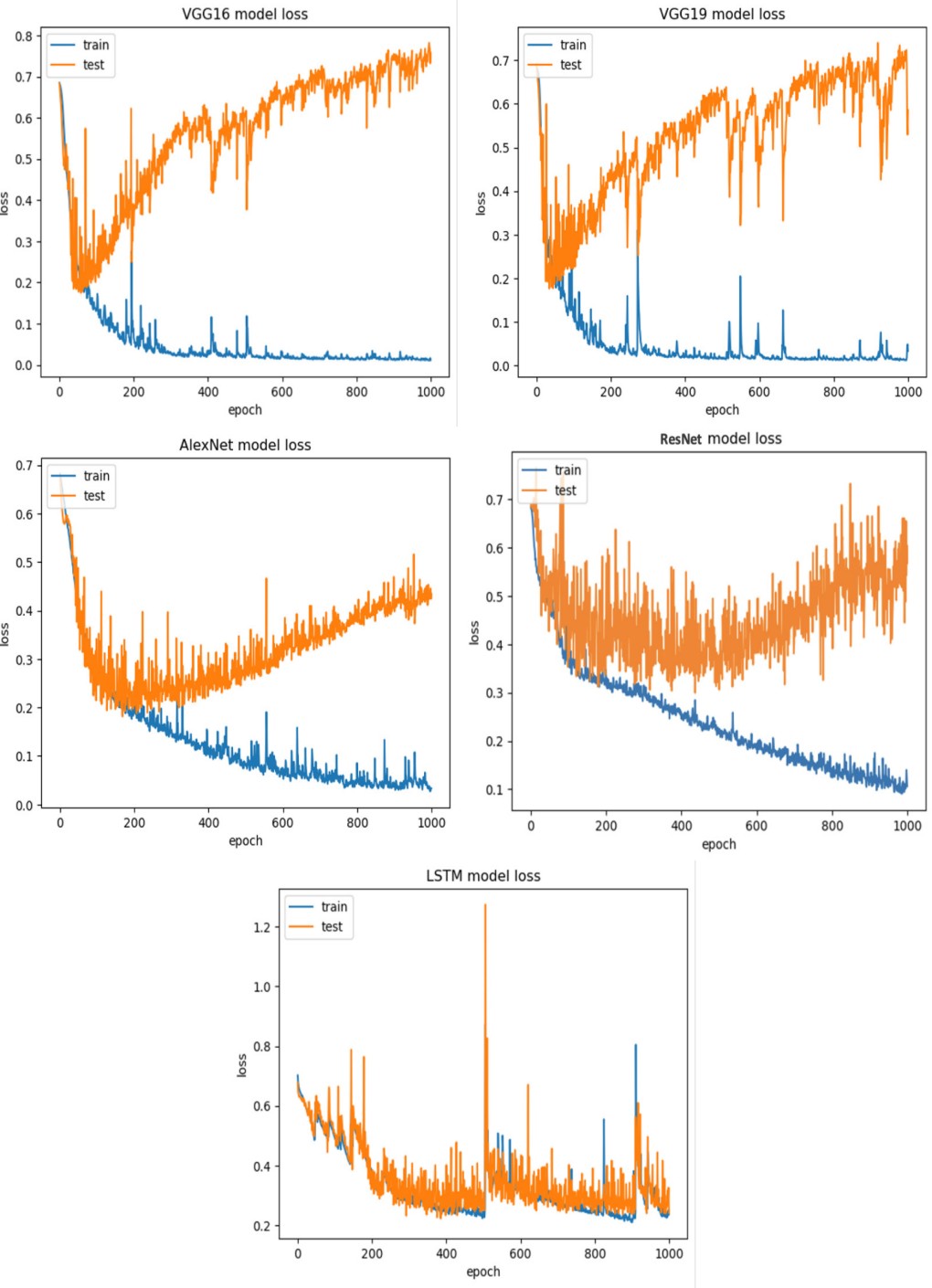

**Figure 13  Viszkok et al.'s dataset (*Viszkok, Hegedus & Ferenc, 2021*) results using five deep learning architectures.**

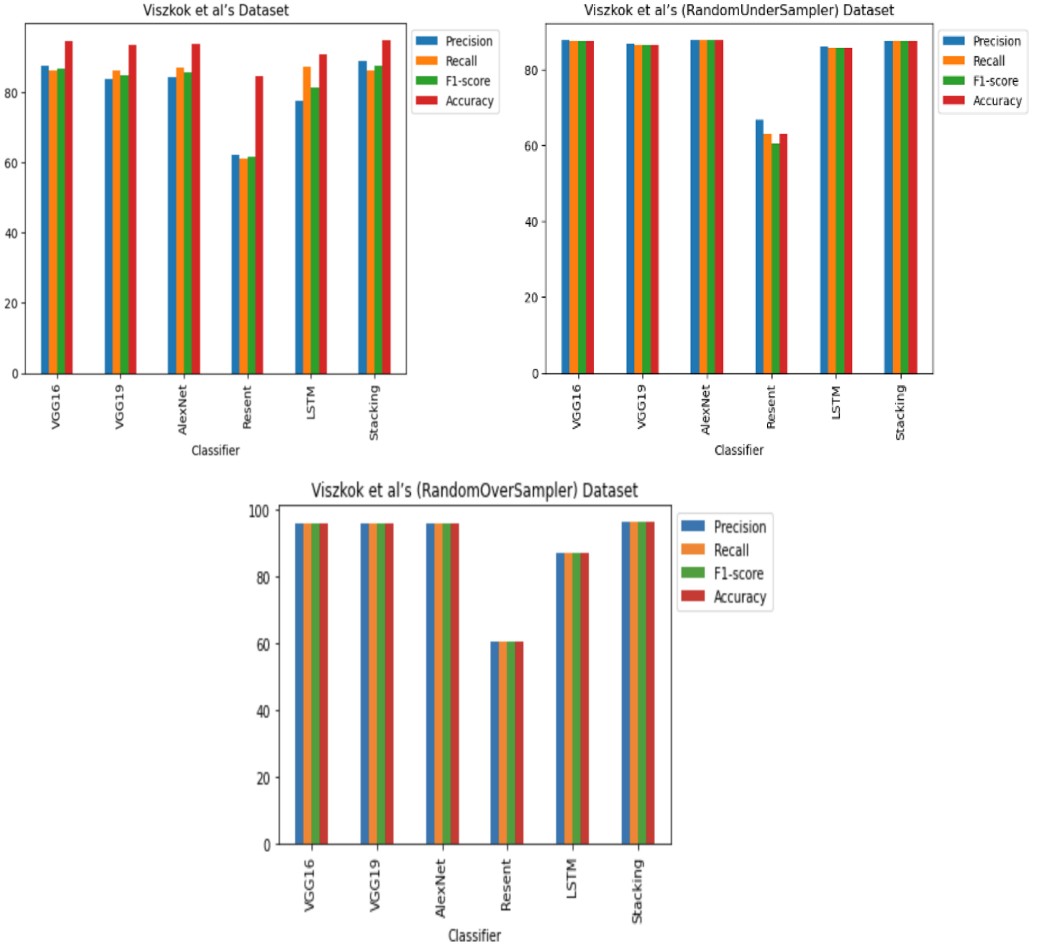

**Figure 14** Viszkok et al.'s dataset (*Viszkok, Hegedus & Ferenc, 2021*) results using deep learning architectures.

types of code vulnerabilities, instead of binary code vulnerabilities, using deep learning approaches. The dataset, such as in Table 6 used for vulnerability code detection is comprised of code fragments with different programming constructs. The dataset includes 7 classes, such as 230,094 non-vulnerability functions, 9,952 pointer usage vulnerabilities, 10,440 input validation bypass vulnerabilities, 13,603 vulnerabilities associated with the exposure of sensitive information, 7,285 API function call vulnerabilities, 3,475 array usage vulnerabilities and 10,926 arithmetic expression vulnerabilities.

Using combined models enables takes advantage of the strengths in each architecture, resulting in an overall performance enhancement. However, the effectiveness of this approach depends on a number of factors, including the specific architectures chosen, the dataset's characteristics, and the hyperparameters assigned to each model. Before they are combined, the performance of each model was individually assessed. The evaluation involves dividing the dataset into 67% training and 33% testing sets, ensuring reliable

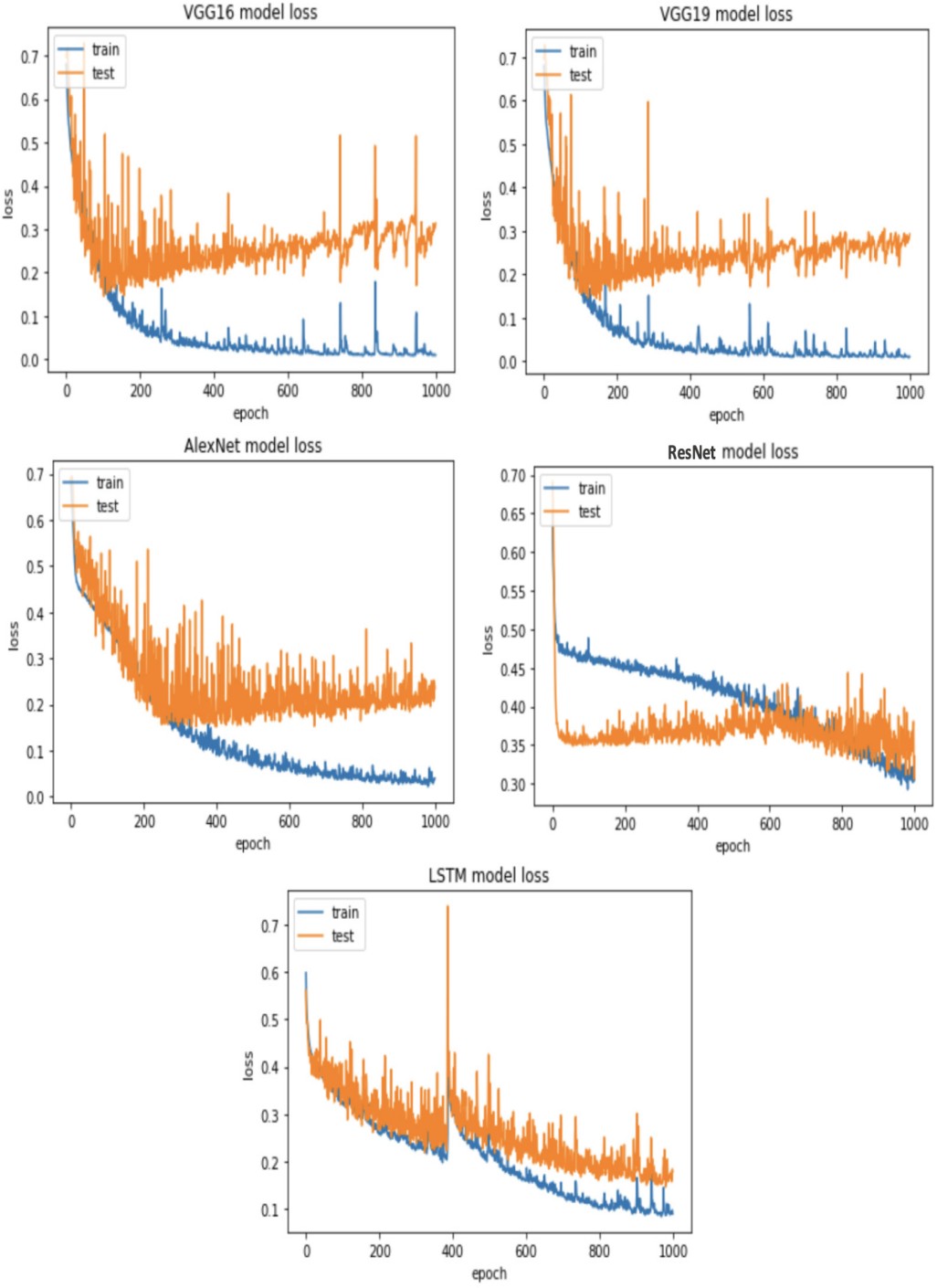

**Figure 15  Apache Tomcat dataset results using five deep learning architectures.**

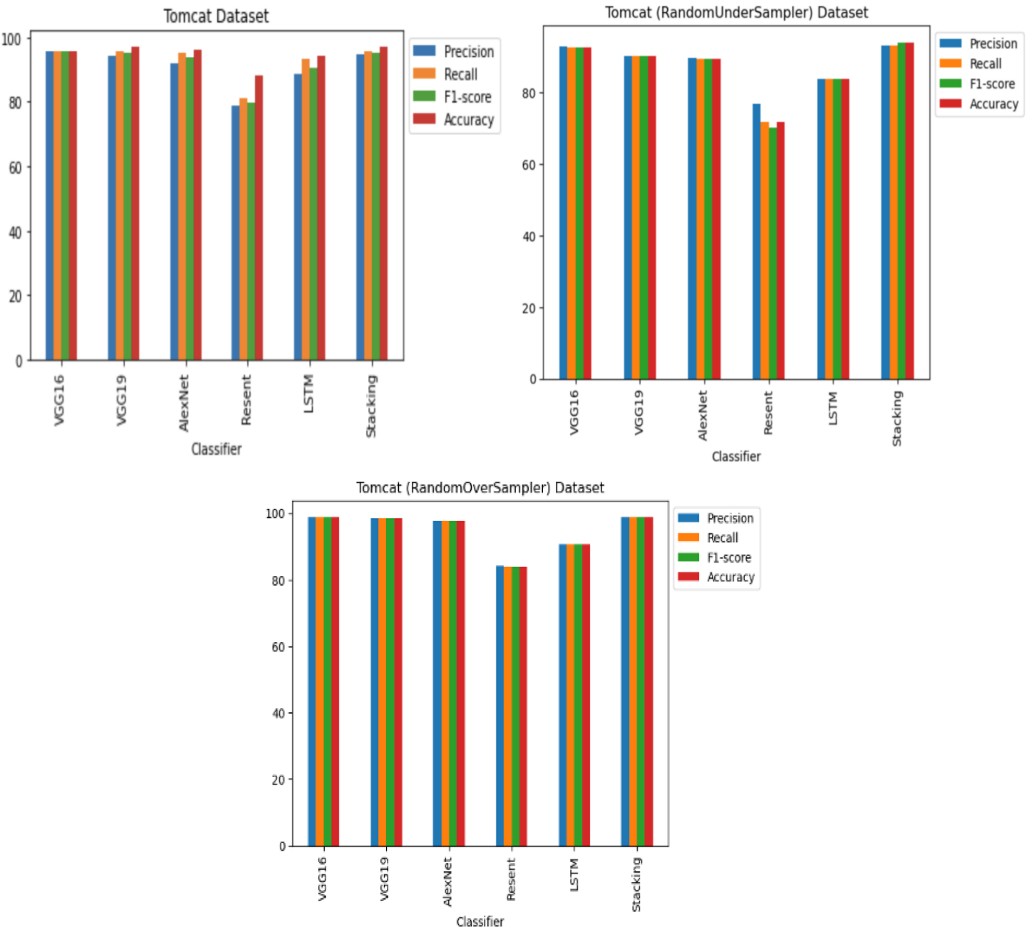

**Figure 16** **Apache Tomcat dataset results using deep learning architectures.**

evaluation. Furthermore, the stacking procedure's hyperparameters were optimized to improve performance.

Figure 17 shows that the stacking CNN architectures VOLUME 4, 2016 performed better than the VGG16, VGG19, AlexNet, ResNet and LSTM architectures. Stacking CNN architectures achieved 73.17% f1-score, 72.97% precision, 73.51% recall and 71.55% accuracy scores. AlexNet achieved the second highest accuracy, at 71.30% and VGG16 achieved 68.31% accuracy. Lower precision was achieved by the other CNN classifiers, recall, f1-score, and accuracy scores rather than the stacking CNN classifiers, AlexNet and VGG16 classifiers.

### Performance comparison with other State-of-the-art approaches

A performance of a proposed stacked CNN model was performed in JavaScript and its vulnerabilities compared to other state-of-the-art methods in a similar class as shown in Tables 7, 8 and 9.

A comparative analysis of various vulnerability JavaScript code detection approaches using machine learning models was also conducted with individual CNN models such as

**Table 6  Types of vulnerabilities (multi-classes) dataset (*Ganesh, Ohlsson & Palma, 2021*).**

| Dataset | Total of functions | Pointer usage | Input validation bypass | Exposure of sensitive information | API function call | Array usage | Arithmetic expression | Non-Vulnerability |
|---|---|---|---|---|---|---|---|---|
| Multi-classes's dataset | 285,775 | 9,952 | 10,440 | 13,603 | 7,285 | 3,475 | 10,926 | 230,094 |

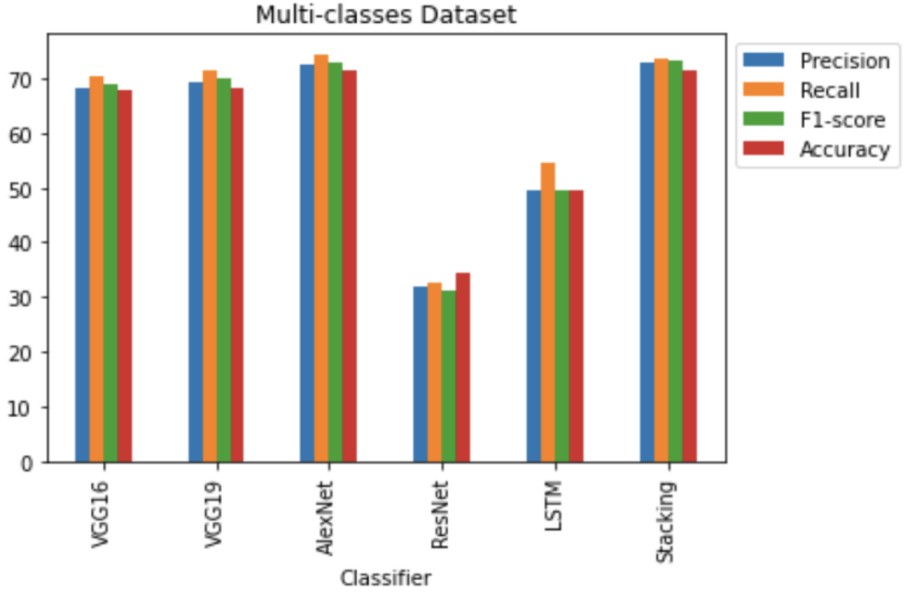

**Figure 17  Multi-classes dataset results using five deep learning architectures.**

**Table 7  Evaluation results of software metrics-based models according *Ferenc et al.*'s (*2019*) dataset using only static metrics.**

| Approach | Precision (%) | Recall (%) | f1-score (%) |
|---|---|---|---|
| DNNs | 87.34 | 59.96 | 71.11 |
| DNNc | 91.06 | 57.89 | 70.78 |
| RF | 93.11 | 57.82 | 71.34 |
| KNN | 90.88 | 65.91 | **76.40** |
| Linear regression | 84.31 | 15.44 | 26.10 |
| Logistic regression | 75.30 | 21.19 | 33.07 |
| SVM | **95.29** | 51.40 | 66.78 |
| Tree | 73.66 | **69.72** | 71.63 |
| Bayes | 22.38 | 11.70 | 15.36 |
| Our stacking CNN classifiers | 87.52 | **80.57** | **83.55** |
| Our Stacking CNN classifiers(RUS) | 81.59 | **81.48** | **81.46** |
| Our stacking CNN classifiers(ROS) | 94.46 | **94.44** | **94.44** |

**Notes.**
The bold values are the best results among other classifiers.

**Table 8** Evaluation results of software metrics-based models according to Viszkok et al.'s dataset (*Viszkok, Hegedus & Ferenc, 2021*) using both process and static metrics.

| Approach | Precision (%) | Recall (%) | F1-score (%) |
|---|---|---|---|
| SDNN | 94.2 | 69.3 | 80.8 |
| CDNN | 93.1 | 71.4 | 80.8 |
| RF | 96.7 | 72.8 | 83.1 |
| KNN | 97.0 | 63.9 | 77.0 |
| Linear regression | 91.0 | 28.5 | 43.5 |
| Logistic regression | 82.4 | 34.6 | 48.7 |
| SVM | 96.8 | 57.1 | 71.8 |
| DT | 90.9 | 75.3 | 82.4 |
| NB | 27.8 | 12.0 | 16.7 |
| Our Stacking CNN classifiers | 88.84 | **86.05** | **87.37** |
| Our stacking CNN classifiers(RUS) | 87.38 | **87.38** | **87.38** |
| Our stacking CNN classifiers(ROS) | **97.06** | **97.05** | **97.05** |

**Notes.**
The bold values are the best results among other classifiers.

**Table 9** Stacking classifier performance in Apache Tomcat (undersampled).

| Approach | Classes | Accuracy | Precision | Recall | F1-Score |
|---|---|---|---|---|---|
| Stacking machine learning classifier (*Ganesh, Palma & Olsson, 2022*) | Vulnerable | 77 | 41.5 | 87.2 | 56.24 |
| | Severity | 66.5 | 66.5 | 66.5 | 66.5 |
| | Title | 10.6 | 10.6 | 10.6 | 10.6 |
| Our approach (stacking CNN classifiers) | Vulnerable | **97.71** | **96.13** | **95.67** | **95.90** |
| | Severity | **94.72** | **79.96** | 71.92 | **73.95** |
| | Title | **93.14** | 52.30 | 52.22 | 49.08 |
| Our approach (stacking CNN classifiers) (RUS) | Vulnerable | 94.04 | 93.09 | 93.04 | 94.04 |
| | Severity | 73.05 | 72.62 | **73.05** | 72.52 |
| | Title | 74.24 | **60.13** | **59.91** | **57.30** |

**Notes.**
The bold values are the best results among other classifiers.

VGG16, VGG19, AlexNet, ResNet and LSTM. The framework was also compared using stacking CNN classifiers with individual CNN models. The experiments were carried out on distinct datasets: *Ferenc et al.*'s (*2019*) dataset, Viszkok et al.'s dataset (*Viszkok, Hegedus & Ferenc, 2021*), and the Apache Tomcat dataset, which included diverse vulnerability types. To ensure an appropriate data distribution, the dataset was divided into training, validation, and test sets, comprising 70%, 15%, and 15% of the samples, respectively. During evaluation, metrics such as accuracy, precision, recall, and F1-score were used for assessing detection approach activity. Ference et al.'s imbalanced dataset results using static metrics shows that the stacking CNN deep learning approach achieved an F1-score of 83.55%, precision of 87.52% and recall of 80.57% which is an improvement on state-of-the-art approaches based on F1-score as shown in Table 7. The stacking CNN deep learning with

random under sampler approach achieved an F1-score of 81.46%, precision of 81.48% and recall of 81.59%. The stacking CNN with random over sampler achieved an F1-score of 94.44%, precision of 94.46% and recall of 94.44%. The SVM classifier achieved a recall of 95.29% which based on precision is the best approach; however, it achieved an F1-score of 66.78% which is worse that the approach based on the F1-score and recall.

The assessment of *Viszkok, Hegedus & Ferenc*'s (*2021*) imbalanced dataset considers both process and static metrics, revealing the impressive performance of the stacking CNN deep learning approach, which achieved an F1-score of 87.37%, precision of 88.84%, and recall of 86.05%. These results improve upon other methods, as shown in Table 8. The highest accuracy, precision, recall and F1-score results for each class is shown in bold. In addition, the stacking CNN deep learning with random under-sampling approach had an F1-score, precision, and recall of 87.38%. Conversely, the stacking CNN with random over-sampler approach demonstrated exceptional performance, achieving an F1-score of 97.05%, precision of 97.06%, and recall of 97.05%. These findings confirm the stacking CNN with random over-sampler approach as the highest achieving of all the methods, with the highest F1-score, whereas, the stacking CNN deep learning and stacking CNN random under-sampler approaches also outperformed other methods. However, the linear regression algorithm displayed the lowest performance metrics. The stacking CNN random under-sampler approach fell in between, with slightly better performance than the CNN deep learning approach but not reaching the level of the stacking CNN random over-sampler approach. The findings reveal the effectiveness of the stacking CNN with random over sampler approach for vulnerability JavaScript code detection. The results are promising and indicate the potential for this approach to efficiently identify vulnerabilities in real-world applications.

The performance of the approach with stacking machine learning classifiers was compared using the Apache Tomcat dataset, which is comprised of three categories of vulnerability types: Vulnerable, Severity, and Title, as shown in Table 8 . When evaluating the vulnerable class, the stacking CNN deep learning model with random under sampler achieved an F1-score of 94.04%. For the Severity class, this approach achieved an F1-score of 72.52%, and for the Title class, it achieved a score of 57.3%. Whereas the stacking CNN deep learning model without random under sampler achieved an F1-score of 95.90% for the vulnerable class, 73.95% for the Severity class, and 49.08% for the Title class. Comparatively, the stacking machine learning classifiers (*Ganesh, Palma & Olsson, 2022*) achieved lower performance scores. In particular, for the vulnerable class they achieved an F1-score of 56.24%, for the Severity class, they achieved 66.50%, and for the Title class, they scored 10.60%. Based on these results, this approach excels in vulnerability software detection across different classes (Vulnerable, Severity, Title) using the Apache Tomcat dataset.

## DISCUSSION OF CHALLENGES USING DEEP LEARNING

This section discusses the challenges and solved gaps in the use of deep learning in order to detect vulnerability JavaScript. Detecting JavaScript vulnerabilities using deep learning

techniques like stacked convolutional neural networks (CNNs) is an advanced and evolving field. Although such approaches are promising, there are several knowledge gaps that were addressed. A significant challenge in using deep learning for vulnerability detection is the lack of availability of labeled data, therefore different vulnerability JavaScripts were combined. Many security datasets have class imbalance issues where most data points represent benign code, therefore, under sampling and oversampling data algorithms were used. In addition, deep learning models can effectively generalize across different frameworks and libraries. Models can struggle to adapt to variations in code structure and patterns. Vulnerability detection models, including CNNs, can also be subject to adversarial attacks. Further research is required to make these models more robust against attacks designed to evade detection such as ensemble or stacked CNNs. Combining stacked CNNs with other machine learning techniques, such as ensemble methods, could also potentially improve detection performance and robustness. An assessment of the real-world impact of using deep learning and stacked CNNs for JavaScript vulnerability detection was conducted in terms of reducing vulnerabilities in web applications and preventing cyberattacks.

## THREAT TO VALIDITY

The data collection process could be potentially inaccurate as it relied on a manual evaluation of additional candidate commits gained from issue comments (*Viszkok, Hegedus & Ferenc, 2021*).

The results may vary depending on the features used for the analysis. If these features cannot capture the essential characteristics of JavaScript code, then the approach may not detect some vulnerabilities or could result in false positives. The training dataset's quality and quantity might impact on the performance of the model. A small or skewed dataset may not sufficiently represent the various vulnerabilities discovered in JavaScript code. The datasets used are enough for model building, and were combined with different datasets for JavaScript to construct a model to identify various types of JavaScript functions that are vulnerable. This is the first known application of stacking CNNs to identify various types of vulnerable JavaScript functions. The programming methodologies or technologies that create JavaScript code may potentially have an effect on the performance of the algorithm. For example, the features that performed well for older versions of JavaScript may be irrelevant to the newer versions, or new vulnerabilities or coding practices may emerge that the model does not capture.

According to privacy issues, we can use blockchain code (*Alamer, 2024*) and *Hakak et al. (2021)* to solve the privacy for vulnerability JavaScript and use a federated learning (FL) system *Alamer (2023)* which is considered as a decentralized ML/DL paradigm that allows IoT devices to train their data locally and only share their trained model with the CC-server without it participating in their source data, thus offering a more privacy-conscious approach.

# CONCLUSIONS

Web-based system security for web applications has recently become more important because of the increase in cyberattacks. The security of these systems could be at risk from weaknesses in JavaScript functionalities. Thus, further research has gone into developing methods to find and identify vulnerabilities. One of these techniques involves stacking convolutional neural networks with a random under sampler and random over sampler in one dimension (*1D*). This is a proven method for identifying vulnerable JavaScript functions. Using CNNs with a *1D* random under sampler and random over sampler has significantly increased the classification accuracy of vulnerable functions.

An objective of this study was to assess the performance of various machine learning classifiers to identify software vulnerability code, considering different dataset scenarios, including imbalanced, under-sampled, and over-sampled cases. Comprehensive training and testing across multiple classifiers was conducted using a diverse collection of code samples classified as either vulnerable or nonvulnerable.

These findings highlight the effectiveness of different classifiers in this important task. Notably, Xg-Boost, Bagging, Random Forest (RF), and stacking classifiers demonstrated better performance for all three datasets and cases. In addition, it is significant to select the correct CNN architecture based on dataset characteristics and class distribution. Stacking, VGG16, VGG19, and AlexNet architectures consistently showed their effectiveness across various scenarios, demonstrating their potential for improving vulnerability software detection.

This study contributes to the field of software security and the detection of vulnerability code, laying the foundation for selecting appropriate CNN models and preprocessing techniques in real-world applications. There is also the potential for further refinement and exploration of these models, which could improve accuracy and dependability in the identification of software vulnerabilities. One question demonstrates the benefits of utilizing stacking CNN architectures in the context of vulnerability code detection. When individual CNN models encounter limitations in situations that involve relatively small datasets and challenges in feature extraction, ensemble methods present a promising answer. Additional research, adjustment, and the examination of hybrid models, which may offer ways by which to enhance the accuracy of vulnerability detection.

The final question demonstrates the potential of using a combination of diverse CNN architectures for multi-class code vulnerability detection. This adds to the research field by demonstrating the effectiveness of deep learning techniques in addressing a complex problem space with real-world implications. Future research could potentially refine this approach, increasing the accuracy and dependability of code vulnerability detection in dynamic and evolving software environments.

The stacked CNN method achieved a detection rate of 98%, highlighting the method's efficacy in identifying vulnerable JavaScript methods. To marks the first known endeavor of its kind to leverage deep learning techniques for the identification of various code vulnerabilities. This research identifies susceptible JavaScript functions by using a combination of CNNs with a *1D* random under-sampler and random over-sampler,

offering essential insights into improving the security of web-based applications and systems against potential threats. By detecting vulnerabilities before they can be exploited by criminals, the risks associated with cyber-attacks can be mitigated and sensitive data protected from potential compromise.

### Funding

The Deputyship for Research & Innovation, Ministry of Education in Saudi Arabia funded this research work through the project number ISP-2024. The funders had no role in study design, data collection and analysis, decision to publish, or preparation of the manuscript.

### Grant Disclosures

The following grant information was disclosed by the author:
The Deputyship for Research & Innovation, Ministry of Education in Saudi Arabia: ISP-2024.

### Competing Interests

The author declares there are no competing interests.

### Author Contributions

- Abdullah Sheneamer conceived and designed the experiments, performed the experiments, analyzed the data, performed the computation work, prepared figures and/or tables, authored or reviewed drafts of the article, and approved the final draft.

### Data Availability

The data is available in the Supplementary Files and at:
- JavaScript Vulnerability Dataset: http://www.inf.u-szeged.hu/ferenc/papers/JSVulnerabilityDataSet/
- Snyk Vulnerability Database: https://security.snyk.io/
- MDPI Data 2022: https://github.com/palmafr/MDPIData2022/tree/main/datasets
The code is available in the Supplemental File.

### Supplemental Information

Supplemental information for this article can be found online at http://dx.doi.org/10.7717/peerj-cs.1838#supplemental-information.

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
