# Peer review of "Vulnerable JavaScript functions detection using stacking of convolutional neural networks"

_PeerJ Computer Science, doi:10.7717/peerj-cs.1838_

## Round 0.1 · original submission · Major Revisions

Dear Authors,

Reviewers suggested significant changes in the manuscript, however, one of the reviewers suggested that the paper needs comprehensive descriptions of the methods used. In the current form, it is very difficult for readers to replicate the study. I also have a few suggestions and queries. You are required to thoroughly read the comments of each of the reviewers and Editor's comments, and revise your manuscript accordingly.

1) There are some repetitive sentences in the abstract. I would recommend dropping the lines "By using an ensemble of CNNs, the proposed method can effectively identify vulnerable code in JavaScript programs, and the approach can be used to enhance the security of web-based systems and applications. This research highlights the significance of using advanced techniques to detect potential vulnerabilities and protect systems against cyber-attacks, contributing to the development of more secure online environments."

2) Write introductory lines related to problem statements before the subheading "An Outline of Vulnerabilities contained within JavaScript" (line no. 33). I feel that the "Background" should be first, then the outline of vulnerabilities contained with JavaScript, then engine architecture, etc.

3) The literature section "Prior Research" is very short, and needs to be extended, and research gaps should be mentioned properly.

4) Line 261, the statement "The dataset and algorithm used in this study were the same as those used by Viszkok et 262 al. Viszkok et al. (2021) and Gyimesi et al. Gyimesi (2017)." - if both dataset and algorithm were adopted from published literature, what is the novelty of this paper? Many of the sentences are quite confusing and unclear. Pl thoroughly proof read and check the sentences. Further, in line 353, "In this research, we utilized various publicly available datasets, including the Apache Tomcat 354 dataset Tom (2023), shown in Table 1, as well as datasets from the Node Security Project Ferenc 355 et al. (2019), the Snyk platform Viszkok et al. (2021), and code-fixing patches from GitHub." - how description about datasets at two different places are conflicting? It must be corrected.

5) Fig. 14 is not clearly visible. Improve its image quality.

6) Line 493, "Performance comparison with other State-of-the-art approaches", here performance of proposed stacked CNN has not been compared with other state-of-the-art similar class of methods, rather it has been compared with individual CNN models such as VGG16, VGG19, AlexNet, ResNet and LSTM. Authors need to compare the proposed framework with other ensembled/stacked based or any other similar category of methods.

7) THREAT TO VALIDITY section is unclear. Authors need to shorten this section, and make it more understandable.

8) There are typos and redundancies across the paper. e.g. CNN has been abbreviated at several places, Line 88, 91, 94, 277, 293, and so on.

9) The provided reference list is not up to date. Cite some relevant and current papers of 2023, 2022, etc.

**Language Note:** The review process has identified that the English language must be improved. PeerJ can provide language editing services - please contact us at [email protected] for pricing (be sure to provide your manuscript number and title). Alternatively, you should make your own arrangements to improve the language quality and provide details in your response letter. – PeerJ Staff

Reviewer 1 ·

Basic reporting

Manuscript should be reread thoroughly as at various places visible ambiguity is present while describing number of datasets used and number of models used in the experimentation.

More background work should be done to establish the need of undertaken work.

Experimental design

Experimental design is clearly described.
Research questions are relevant and well defined.
The response to these questions should be made more understandable may be by incorporating some tables for results or figures to more clearly establish conclusions given in the response to research questions.

Validity of the findings

Novelty of the undertaken study is not clear. More emphasis should be given on why the proposed work is important.
The contributions made by authors do not seem to be sufficient in its current form for publication.

Additional comments

The manuscript covers important and relevant research issue but Novelty of the work is ambiguous. Also the manuscript lacks consistency in some places.

Reviewer 2 ·

Basic reporting

The language and references sometimes need to be more specific and clear. For instance, in line 282: "In this study, we utilized the VGG16 model by training it with pre-trained weights," it is unclear if they used a pre-trained model or something else. Figures are references out of order. For instance, they first refer to Figure 8, and they Figure 7.

There are notable issues with the article's structure and referencing. The references need more specificity and often lead to clarity. For example, on line 282, it needs to be clarified if the authors used a pre-trained model. Furthermore, referencing figures must be more consistent (for instance, on line 285). The text sometimes refers to the wrong figure, as seen in Figure 7 and Figure 8. To enhance the paper's clarity, these issues must be addressed.

The representation of the results needs improvements. Figure 8 shows the performances of different methods, but there are better choices of graphs. Moreover, the graph gets pixelated if one zooms in. I recommend using vector graphics instead.

Experimental design

The research's primary focus is the development of a stacked CNN architecture for identifying functional vulnerabilities, aligning with the journal's Aims and Scope. The research question is defined, albeit with room for improvement in terms of clarity and depth. Additionally, there is a need for more explicit communication regarding the existing knowledge gaps in this area.

The investigation demonstrates a rigorous technical and ethical standard. However, the paper needs comprehensive descriptions of the methods used, making it difficult for readers to replicate the study. More detailed information regarding the architecture and hyperparameters of the employed models is necessary, as previous studies have shown that hyperparameter tuning optimization could drastically impact SE tools' performance [1].



[1] Tantithamthavorn, Chakkrit, et al. "Automated parameter optimization of classification techniques for defect prediction models." Proceedings of the 38th international conference on software engineering. 2016.

Validity of the findings

The paper needs to assess the impact and novelty of its findings. Unfortunately, there is no replication package to check the code, review their code and reproduce their findings.
Conclusions need to be more precisely linked to the original research question and confined to supporting results. The current lack of connection between conclusions and research questions may need to be clarified for readers.

Based on Figure 8, Random Forest performs as well as the stacked model, so why should one choose the stacked model, which I believe is more computationally expensive?

Additional comments

Overall, I believe the authors should address these issues and resubmit the paper.

---

## Round 0.2 · Minor Revisions

One of the reviewers suggested further modifications. Address these concerns and resubmit.

Reviewer 1 ·

Basic reporting

Article is well written.

Experimental design

Authors have modified the manuscript as suggested.

Validity of the findings

Findings are well stated and work is relevant.

Additional comments

Work done by authors is concerete and is relevant.

Reviewer 2 ·

Basic reporting

Basic Reporting:
1. Improvements Acknowledged: The revisions made to the manuscript have notably enhanced its clarity and coherence. The effort put into addressing the feedback from the initial review is commendable.

Experimental design

Experimental Design:
2. Clarity of Research Goals: The goals of the research are now articulated more clearly, which significantly aids in understanding the scope and intent of your study.
3. Concerns Regarding Data Sources: The use of a dataset from an Arxiv paper raises concerns about the validity of the dataset, given that it has not undergone formal peer review. This issue should be addressed to ensure the reliability of your findings.
4. Replicability Issue: A major concern is the absence of the CSV file for the dataset used in your code. This omission significantly hinders the ability to replicate your work. To verify the replicability of your study, it is imperative that you provide the necessary CSV files. The code appears to be in order, but without the dataset, a critical aspect of your research remains unverifiable.

Validity of the findings

Validity of the Findings:
5. Dataset Availability: Although the paper is well-structured, the lack of access to the dataset precludes a thorough validation of your findings. Providing the dataset is crucial for assessing the validity of your work.

Additional comments

Minor Issues:
There were few formatting errors which I mention two of such samples samples below. Please recheck the paper in the revision.
Ex1. Two datasets were in this study from the Node Security ProjectFerenc et al. (2019) and ..
Ex2. On page 15: the formatting of figure references in the sentence “…in Figures 8,9 and 10, and…” requires a minor correction to “…in Figures 8, 9, and 10, and…” for consistency and adherence to editorial standards.

---

## Round 0.3 · accepted · Accept

The author has revised the manuscript. The manuscript may be accepted in its current form.